# Diffusion Language Models Can Perform Many Tasks with Scaling and Instruction-Finetuning

## Abstract

The recent surge of generative AI has been fueled by the generative power of diffusion probabilistic models and the scalable capabilities of large language models. Despite their potential, it remains elusive whether *diffusion language models* can solve general language tasks comparable to their autoregressive counterparts. This paper demonstrates that scaling diffusion models *w.r.t.* data, sizes, and tasks can effectively make them strong language learners. We build competent diffusion language models at scale by first acquiring knowledge from massive data via masked language modeling pretraining thanks to their intrinsic connections. We then reprogram pretrained masked language models into diffusion language models via diffusive adaptation, wherein task-specific finetuning and instruction finetuning are explored to unlock their versatility in solving general language tasks. Experiments show that scaling diffusion language models consistently improves performance across downstream language tasks. We further discover that instruction finetuning can elicit zero-shot and few-shot in-context learning abilities that help tackle many unseen tasks by following natural language instructions, and show promise in advanced and challenging abilities such as reasoning.

## 1 Introduction

Recent advances in generative modeling have led to remarkable progress in the field of generative AI. In domains of continuous signals, diffusion probabilistic models have shown great success in rendering photorealistic images (Rombach et al., 2021; Ramesh et al., 2022) and synthesizing high-quality audio (Kong et al., 2020) through iterative denoising, outperforming GANs and autoregressive models, and even contributing to the surge of AI art. The story is different in the domains of discrete signals comprising symbolic sequences such as natural languages, where autoregressive large language models (large LMs or LLMs, Brown et al., 2020) have dominated the scene, delivering impressive generalist language abilities in language understanding and generating human-like texts, and can even follow natural language instructions to perform unseen tasks.

The revolutionized generative abilities of diffusion models, manifested in image generation and speech synthesis, give the promise of a strong alternative to autoregressive language models for several favorable reasons, including (1) global receptive field *vs.* one-sided context, and (2) non-autoregressive drafting-then-revising manner *vs.* restrictive unidirectional generation/autoregression. Hence, an intriguing question arises: *can diffusion models speak languages well?* This question is in turn asking about the *scalability* of diffusion language models, which can be further boiled down into the following specific research questions regarding the three key ingredients of the success of large-scale language models, *i.e.*, data, model sizes, and tasks:

RQ 1. *On scaling data.* Acquiring general knowledge via self-supervised pretraining from massive unlabeled data plays a crucial role in the success of the modern NLP paradigms (Radford et al., 2018; Devlin et al., 2018), hence it is also of importance to enable diffusion language models to learn from massive data. ***Can diffusion language models leverage knowledge from large-scale data?***

RQ 2. *On scaling model sizes.* It has been widely observed that the larger the model size, the more competent the language models become. ***Can enlarging diffusion language models effectively improve downstream tasks?***

RQ 3. *On scaling tasks.* What makes LLMs most attractive is they can tackle new tasks that they were never exposed to during training by following natural language instructions with little

Figure 1: Overview. (A) Comparative illustration of language model (LM) paradigms, *i.e.*, autoregressive LMs *vs.* diffusion LMs. (B) Overall illustration of the proposed approach wherein massively pretrained masked LMs are reprogrammed to diffusion LMs via *generative surgery*.

to no demonstrations. ***Can scaled diffusion language models exhibit general zero-shot and few-shot in-context learning capabilities to generalize to unseen tasks?***

Nevertheless, building diffusion language models at scale is non-trivial. Previous efforts mostly still fall short of satisfactory generation quality, and the scalability remains largely unexplored. Several studies attempted to adapt continuous diffusion models to discrete domains by embedding discrete symbols into continuous surrogates (Li et al., 2022; Gong et al., 2022; Gao et al., 2022; Ye et al., 2023). However, a significant performance gap persists due to the *pitfall of discreteness* (Ye et al., 2023), which renders Gaussian perturbation ineffective in providing training signals to learn on discrete tokens. Discrete diffusion models, which directly operate in the discrete space, appear well-suited for sequence learning (Hoogeboom et al., 2021; Austin et al., 2021). However, they have long-standing struggles applying to more complex and practical scenarios (typically with large vocabulary) like natural languages. Very recently, *reparameterized discrete diffusion models* (RDM, Zheng et al., 2023a) has made substantial progress on representative benchmarks like machine translation. In addition, He et al. (2023) demonstrated DiffusionBERT, a discrete diffusion model finetuned from pretrained masked language models (MLMs, Devlin et al., 2018). Likewise, Zheng et al. (2023b) also showed that the generative ability can be unleashed from pretrained protein MLMs (Rives et al., 2019) for designing protein amino acid sequences in a diffusion-like iterative refinement fashion. Despite such promising progress, the scalability of diffusion language models remains elusive.

In this paper, we aim to advance diffusion language models by exploring their scalability *w.r.t.* data, model sizes, and tasks. We first demonstrate the intrinsic connection between masked language models and discrete diffusion models, which permits us to treat pretrained masked language models of various scales as pretrained diffusion language models, without the need for expensive learning from scratch. We then reprogram pretrained masked language models into diffusion language models via *diffusive adaptation*, where task-specific finetuning and instruction finetuning (Wei et al., 2021) are explored for solving certain targeted downstream tasks or general language problems, respectively.

Based on extensive experiments, we reveal that large-scale diffusion language models can serve as strong sequence generative models, exhibiting competitive performance as compared with autoregressive language models. Scaling up diffusion language models helps achieve improved performance across a wide range of tasks, from translating across languages to summarizing documents. By leveraging instruction finetuning, we can further elicit zero-shot and few-shot abilities for diffusion language models to tackle unseen tasks by following natural language instructions. Notably, diffusion language models demonstrate promising structured reasoning behaviors thanks to their flexible non-autoregressive generation order. Nevertheless, their capacity to tackle complex reasoning tasks remains an ongoing challenge awaiting resolution.

To sum up, we hope that our explorations provide valuable insights into the scalability of diffusion language models and their potential as a viable alternative in tackling generative language tasks across the board.

## 2 PRELIMINARIES: DIFFUSION MODELS FOR SEQUENCE GENERATION

Language processing tasks can be unified as sequence-to-sequence problems (Raffel et al., 2020), modeling the conditional distribution $p_\theta(\boldsymbol{x}|\boldsymbol{c})$, where $\boldsymbol{x} = (\boldsymbol{x}^{[1]}, \boldsymbol{x}^{[2]}, \ldots, \boldsymbol{x}^{[N]})$ is a target sequence composing $N$ tokens and $\boldsymbol{c}$ is the given context. For example, we may want to generate responses $\boldsymbol{x}$ conditioned on the prompt $\boldsymbol{c}$, or it can be unconditional generation if no context is provided (*i.e.*, $\boldsymbol{c} = \phi$). As a result, one thing we care about is the capability of generative models for sequence data

$\boldsymbol{x}$, *e.g.*, the prevailing autoregressive models or diffusion models. In this section, we provide the necessary background on diffusion-based sequence generative models, where we abuse the notation and use $p_\theta(\boldsymbol{x})$ for both conditional $p_\theta(\boldsymbol{x}|\boldsymbol{c})$ and unconditional $p_\theta(\boldsymbol{x}|\boldsymbol{c} = \phi)$ for brevity. We provide more detailed discussions on relevant literature in §B.

**Diffusion Models** (Sohl-Dickstein et al., 2015) are a class of generative models characterized by a pair of Markov processes, *i.e.*, a forward diffusion process and a backward denoising process. The *forward* process $q(\boldsymbol{x}_{1:T}|\boldsymbol{x}_0) = \prod_{t=1}^{T} q(\boldsymbol{x}_t|\boldsymbol{x}_{t-1})$ gradually perturb the data $\boldsymbol{x}_0 \sim q(\boldsymbol{x}_0)$ into a stationary distribution $q(\boldsymbol{x}_T)$ with $T$ increasingly noisy steps $\boldsymbol{x}_{1:T} = \boldsymbol{x}_1, \boldsymbol{x}_2, \ldots, \boldsymbol{x}_T$. The learned *backward* process $p_\theta(\boldsymbol{x}_{0:T}) = p(\boldsymbol{x}_T) \prod_{t=1}^{T} p_\theta(\boldsymbol{x}_{t-1}|\boldsymbol{x}_t)$, reversely, gradually denoises the samples towards the data distribution. To fit the model $p_\theta(\boldsymbol{x}_0)$ to the data distribution $q(\boldsymbol{x}_0)$, the denoiser model is typically optimized by the variational bound of the negative log-likelihood (Ho et al., 2020):

$$\mathbb{E}_{q(\boldsymbol{x}_0)}\left[-\log p_\theta(\boldsymbol{x}_0)\right] \leq \mathbb{E}_{q(\boldsymbol{x}_{0:T})}\left[-\log \frac{p_\theta(\boldsymbol{x}_{0:T})}{q(\boldsymbol{x}_{1:T}|\boldsymbol{x}_0)}\right] = L_1 + \sum_{t=2}^{T} L_t + \text{const.}, \quad (1)$$

where $L_1 = \mathbb{E}_q\left[-\log p_\theta(\boldsymbol{x}_0|\boldsymbol{x}_1)\right]$ and $L_t = \mathbb{E}_q\left[\text{KL}[q(\boldsymbol{x}_{t-1}|\boldsymbol{x}_t, \boldsymbol{x}_0)\|p_\theta(\boldsymbol{x}_{t-1}|\boldsymbol{x}_t)]\right]$ for $t \in [2, T]$.

In general, diffusion models can be categorized into continuous and discrete diffusion models according to distribution type for data perturbation. Continuous diffusion models with Gaussian perturbation have demonstrated impressive performance in generating continuous signals (Rombach et al., 2021; Ho et al., 2022; Kong et al., 2020) but still struggle with satisfactory generation quality in natural languages (Li et al., 2022; Gong et al., 2022; Gao et al., 2022; Yuan et al., 2022; Ye et al., 2023). A critical challenge herein is the *pitfall of discreteness* (Ye et al., 2023) that makes Gaussian perturbation on embeddings hardly provide effective training signals. In contrast, discrete diffusion models directly operate over the discrete state space of tokens, providing an attractive alternative for generative sequence learning. Therefore in this paper, we explore developing diffusion language models upon discrete diffusion.

**Discrete Diffusion Models** (Hoogeboom et al., 2021; Austin et al., 2021) cover a subset of diffusion models for which transition probabilities between timesteps are discrete distributions. Since the forward diffusion process is applied independently to each token of a sequence $\boldsymbol{x}$, for the sake of brevity, we abuse the notation $\boldsymbol{x}_t$ for arbitrary tokens at diffusion timestep $t$. Formally, $\boldsymbol{x}_t \in \{0, 1\}^{|\mathcal{V}|}$ is a token represented as a one-hot vector, where $\mathcal{V}$ is the vocabulary of all possible tokens. Let $\text{Cat}(\boldsymbol{x}; \boldsymbol{p})$ be a categorical distribution on $\boldsymbol{x}$ with probabilities given by vector $\boldsymbol{p}$ on $|\mathcal{V}| - 1$ dimensional probability simplex, and the forward transition be $q(\boldsymbol{x}_t|\boldsymbol{x}_{t-1}) = \text{Cat}\left(\boldsymbol{x}_t; \boldsymbol{p} = \beta_t \boldsymbol{x}_{t-1} + (1 - \beta_t)\boldsymbol{q}_{\text{noise}}\right)$, where $0 \ll \beta_t < 1$ is the noise schedule controlling the degree of perturbation at timestep $t$, and $\boldsymbol{q}_{\text{noise}}$ is the probability vector of stationary distribution $q(\boldsymbol{x}_T)$, *i.e.*, $q(\boldsymbol{x}_T) = \text{Cat}(\boldsymbol{x}_T; \boldsymbol{p} = \boldsymbol{q}_{\text{noise}})$. In this case, the distribution of corrupted sample $\boldsymbol{x}_t$ given its original data $\boldsymbol{x}_0$ has a closed-form expression:

$$q(\boldsymbol{x}_t|\boldsymbol{x}_0) = \text{Cat}\left(\boldsymbol{x}_t; \boldsymbol{p} = \alpha_t \boldsymbol{x}_0 + (1 - \alpha_t)\boldsymbol{q}_{\text{noise}}\right), \quad (2)$$

where $\alpha_t = \prod_{i=1}^{t} \beta_i$. This shows that the diffusion process is intuitively a convex combination between data and noise where the $\alpha_t$ controls the degree of corruption at different timesteps. In particular, $\alpha_t$ decreases as the timestep increases. With sufficiently large timesteps, we have $\alpha_T \approx 0$, which preserves no information from the data at the end of the diffusion process.

Different stationary distributions $\boldsymbol{q}_{\text{noise}}$ lead to different formulations of discrete diffusion models. One typical design is the *absorbing* diffusion with $q(\boldsymbol{x}_T) = \{1 \text{ if } \boldsymbol{x}_T = \texttt{[MASK]}; 0 \text{ if } \boldsymbol{x}_T \neq \texttt{[MASK]}\}$, where $\texttt{[MASK]}$ is an absorbing state. According to Eq. (2), this formulation results in $\boldsymbol{x}_t$ either being masked or the same as $\boldsymbol{x}_0$, with a masking ratio $(1 - \alpha_t)$. This makes absorbing diffusion resemble masked language models (MLM, Devlin et al., 2018) as He et al. (2023) points out.

**Reparameterized Discrete Diffusion Models** (RDM, Zheng et al., 2023a) reparameterize the backward transition of diffusion language models that reformulates the training objective of discrete diffusion models into

$$L_t = \mathbb{E}\left[-\lambda_{t-1}^{(2)}\left(1 - \mathbb{1}(\boldsymbol{x}_t = \boldsymbol{x}_0)\right)\log p_\theta(\boldsymbol{x}_0|\boldsymbol{x}_t)\right], \quad (3)$$

where $\mathbb{1}(\cdot)$ is indicator function. Under the formulation of absorbing diffusion, Eqn. 5 resembles a weighted MLM objective (Devlin et al., 2018). Zheng et al. (2023a) demonstrate that Eqn. 5 is a more effective training protocol compared to Eqn. 1 for generative discrete diffusion models, showing performance on par with autoregressive LMs (Vaswani et al., 2017) on representative machine translation benchmarks for the first time. In this paper, we use RDM as our primary training objective for building our diffusion language models (see §C for more details).

**Generative Process of Discrete Diffusion Models.** Diffusion models yield new samples by their reverse generative process of iterative denoising. Under the formulation of absorbing diffusion, the denoising process can be characterized in an iterative *mask-predict* manner (Ghazvininejad et al., 2019). Specifically, the starting sequence is initialized by all [MASK] tokens, and in each iteration, some masked tokens are replaced by the model predictions from $p_\theta(\boldsymbol{x}_{t-1}|\boldsymbol{x}_t)$ while some unmasked tokens are remasked, according to specific strategies/schedules (Ghazvininejad et al., 2019; Savinov et al., 2021; Chang et al., 2022; Zheng et al., 2023a). In this paper, we follow Zheng et al. (2023a) to unmask positions with top-$k$ log-probability predicted by $p_\theta(\boldsymbol{x}_0|\boldsymbol{x}_t)$, and mask all the rest position in each denoising step[1].

## 3 SCALING DIFFUSION LANGUAGE MODELS *w.r.t.* DATA, SIZES AND TASKS

Developing diffusion language models that leverage the advantages of both the generative power of both diffusion models and the scalability of large pretrained language models is a promising yet challenging endeavor. The key to the success of the current standard paradigm of large generative language models is acquiring knowledge via massive pretraining and generating in a prompt-response manner for preferable output for many tasks. For diffusion language models, (1) how to benefit from pretraining at scale, and (2) how to best fit the prompt-response paradigm, are the crucial open questions. In this section, we will elaborate on how to empower diffusion language models with knowledge from pretraining of large-scale data as well as model sizes, and extend their generative capabilities for extensive downstream tasks.

### 3.1 KNOWLEDGE ACQUISITION VIA MLM PRETRAINING AT SCALE

The theoretical framework of discrete diffusion models has an intrinsic connection to masked language modeling (MLM), which was discussed in Austin et al. (2021); Gong et al. (2022) and He et al. (2023). Among various types of discrete diffusion models, the *absorbing* diffusion (Austin et al., 2021) resembles a *generalized* masked language modeling, which has been shown to be an effective training objective in pretraining foundation models (Devlin et al., 2018; Liu et al., 2019). Specifically, absorbing diffusion defines a stationary distribution: $q(\boldsymbol{x}_T) = \{1$ if $\boldsymbol{x}_T = $ [MASK]; $0$ if $\boldsymbol{x}_T \neq$ [MASK]$\}$, where [MASK] is an absorbing token. According to Eq. (2), this formulation results in $\boldsymbol{x}_t$ either being masked or the same as $\boldsymbol{x}_0$, with a masking ratio $(1 - \alpha_t)$. Consequently, $\boldsymbol{x}_t = \boldsymbol{x}_0$ if and only if $\boldsymbol{x}_t \neq$ [MASK], which aligns the reparameterized training objective in Eq. (5) exactly with the masked language modeling objective.

This connection allows us to establish diffusion language models by pretraining with MLM objectives from massive raw textual data. We can even treat abundant community-available pretrained MLMs (Devlin et al., 2018; Liu et al., 2019; Conneau et al., 2019) as pretrained diffusion language models, and can depart from them for downstream tasks at a very low cost, bypassing the expensive pretraining stage.

### 3.2 DIFFUSIVE ADAPTATION: REPROGRAMMING PRETRAINED MLMS TO DIFFUSION LANGUAGE MODELS FOR SEQUENCE GENERATION

Existing masked language models are primarily designed to serve as sequence encoders, and are not able to generate sequences by default. Despite their connections to absorbing discrete diffusion, it is non-trivial to naively sample from masked language models through the iterative denoising process of absorbing diffusion. One major reason is that absorbing diffusion generates sampling by iterative applying $p_\theta(\boldsymbol{x}_{t-1}|\boldsymbol{x}_t)$ from complete noise to the final prediction (*i.e.*, ranging gradually from $100\%$ to $0\%$ [MASK] tokens) through different timesteps, whereas vanilla masked language models are only pretrained with a limited and constant masking ratio (*e.g.*, 15%).

In order to elicit the pretrained masked language models' ability for sequence generation, we propose *diffusion adaptation* to eliminate the gap between pretrained masked and diffusion language models, where we further finetune pretrained MLMs with diffusion training objective such that sampling with the denoising process becomes possible. In particular, we follow the reparameterized training and sampling method in RDM (Zheng et al., 2023a) as described in §2. As for model architecture, we

---

[1]See §D for concrete noise schedules, and Zheng et al. (2023a) for the justification of this sampling strategy.

adopt a decoder-only architecture[2] and do not add extra timesteps indicators to the models, similar to He et al. (2023). In this way, our diffusion language model starts as a fully pretrained model without any randomly initialized parameters incorporated. In addition, we incorporate a task-specific length predictor, a common practice in NAR text generation (Gu et al., 2018), to determine the lengths of predicted sequences. We pick its tok-$k$ length predictions for parallel length beam search, where $k$ is referred to as the length beam size. We include more implementation details in §D.

For different purposes, we perform diffusive adaptation for diffusion language models in two ways:

- ***Optimizing specialist capabilities on certain downstream tasks via task-specific finetuning.*** To verify the feasibility of diffusive adaptation, we finetune pretrained masked language models on the specific dataset for each downstream task. Moreover, we further perform finetuning on pretrained models of different scales so as to study the scalability of diffusion language models.

- ***Eliciting generalist capabilities on extensive tasks via instruction finetuning.*** Finetuning on a collection of tasks phrased as instructions (*i.e.*, instruction finetuning) enables language models to better respond to instruction prompts and generalize to unseen tasks (Wei et al., 2021; Chung et al., 2022). Inspired by this, we apply diffusive adaptation to pretrained masked language models by instruction finetuning to study whether diffusion language models can acquire few-shot and zero-shot abilities like autoregressive LLMs.

Both scenarios handle conditional sequence generation tasks from input to output, which require the model to generate target sequences according to the given prompts. To handle these with a decoder-only model, we organize the data in a prompt-response format[3]. During tuning, we only apply the diffusion process to the target response tokens and compute loss on them. During inference, we append the initial fully masked sequences to the prompts and denoise from them.

## 4 EXPERIMENTS

We first introduce our general experimental setups in §4.1. Then we conduct three parts of experiments progressively regarding scaling on data (§4.2), model sizes (§4.3), and the number of tasks (§4.4).

### 4.1 EXPERIMENTAL SETUP

**Model architecture.** Throughout the experiments, we use XLM-RoBERTa (`XLM-R`; Conneau et al., 2019; Goyal et al., 2021) as our foundation models, which is pretrained on CC100 (Wenzek et al., 2020), a multilingual corpus containing 167B tokens of 100 languages, with four model sizes (numbers of non-embedding parameters) at different scales, *i.e.*, 86M, 304M, 2.8B, and 9.7B.

**Data.** We investigate our approach for its specialist ability in respective downstream tasks and generalist ability to solve massive unseen tasks using natural language instructions. The datasets we use to finetune our model are as follows:

*(1) Downstream task datasets.* We evaluate whether our approach can help diffusion language models serve as strong specialized models on multiple representative downstream tasks: (1) IWSLT14[4] for DE→EN translation; (2) WMT14[5] for EN→DE translation; and (3) Gigaword-10K[6] for text summarization.

*(2) Instruction finetuning datasets.* We follow Chung et al. (2022) and finetuned the XLM-R models of different scales with the Flan 2022 Collection (Chung et al., 2022; Longpre et al., 2023) with diffusion training objective in Eq. (5). The Flan 2022 Collection is the publicly available version of the instruction tuning data for Flan-T5 and Flan-PaLM, covering over 1.8K tasks. It combines several multitask learning datasets with instructions (Wei et al., 2021; Sanh et al., 2021; Wang et al., 2022), combined with a few extra chain-of-thought and dialog data.

---

[2]In this paper, the decoder-only architecture, as a counterpart of encoder-decoder architecture, refers to the language model architecture that does not comprise a separate encoder stack to encode contexts/conditions. Under this definition, masked language models (*e.g.*, BERT and XLM-R) are treated as decoder-only models.

[3]A prompt-response formatted example for German-to-English translation ("Vielen dank" - "Thank you"): "Translate the German sentence into English. German: Vielen dank. English: Thank you."

[4]https://wit3.fbk.eu/

[5]http://www.statmt.org/wmt14/translation-task

[6]https://github.com/harvardnlp/sent-summary

Table 1: SacreBLEU (Post, 2018) on IWSLT14 DE→EN and WMT14 EN→DE, and Rouge-L on Gigaword-10k. We use 10 length beams for all the results with length prediction. Results out of (inside) parentheses are obtained with length prediction (oracle target length). "#Params.": Number of non-embedding parameters. "Type": whether the training objective and sampling method are autoregressive (AR, Vaswani et al., 2017) or follow reparameterized diffusion models (RDM, Zheng et al., 2023a). "Pretrained": whether initialized from pretrained models. "†": `Transformer-BASE-IWSLT` has the same architecture as the `Transformer-BASE` in Vaswani et al. (2017) but the dimension of the feed-forward layers is 1024 and the number of attention heads is 4, which is standard practice on this dataset.

|  | Architecture | #Params. | Type | Pretrained | IWSLT14 | WMT14 | Gigaword-10K |
|---|---|---|---|---|---|---|---|
| Encoder-Decoder | `Transformer-BASE-IWSLT`† | 39M | AR | ✗ | 33.30 | - | - |
|  |  | 39M | RDM | ✗ | 32.14 | - | - |
|  | `Transformer-BASE` | 43M | AR | ✗ | - | 26.85 | 10.42 |
|  |  | 43M | RDM | ✗ | - | 26.54 | - |
| Decoder-only | `XLM-R-BASE` | 86M | AR | ✗ | 26.07 | - | - |
|  | `XLM-R-BASE` | 86M | RDM | ✗ | 28.79 (29.12) | 26.09 (26.86) | 10.01 (10.66) |
|  | `XLM-R-BASE` | 86M | RDM | ✓ | 34.10 (35.78) | 26.65 (26.64) | 27.52 (28.83) |
|  | `XLM-R-XXL` | 9.7B | RDM | ✓ | **38.57 (40.65)** | **30.34 (32.81)** | **31.54 (32.57)** |

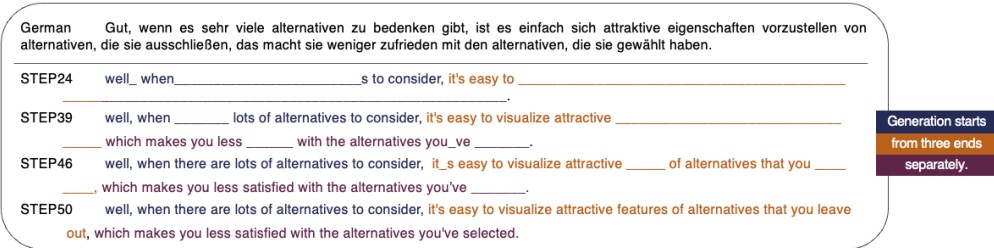

Figure 2: An exemplary generation process on machine translation. Notice that the target translation contains three segments, which are generated simultaneously by the diffusion language model.

## 4.2 DIFFUSION LANGUAGE MODELS BENEFIT FROM LARGE-SCALE DATA

We apply diffusive adaptation on `XLM-R-BASE` (Conneau et al., 2019) model architecture on sequence generation benchmarks to verify the feasibility of our generative surgery and investigate whether diffusion LMs can benefit from large-scale self-supervised learning. We sample the target sequences with 50 steps during inference. For comparison, we include the commonly used encoder-decoder Transformer (Vaswani et al., 2017), and decoder-only autoregressive LMs with the same architecture from scratch as the baselines[7] to help intuitively understand the model performance.

**Diffusive adaptation unlocks the generative ability of pretrained masked language models.** Tab. 1 shows our results on IWSLT14 DE→EN and WMT14 EN→DE translation, as well as Gigaword-10k summarization tasks. The performance of finetuned XLM-R models is competitive or superior to the common encoder-decoder Transformer baseline. As qualitatively shown in Fig. 2, diffusion language models generate fluent and semantically accurate translation[8], further confirming the feasibility of our generative surgery to the pretrained masked language models.

**MLM pretraining at scale benefit diffusion language models.** On both IWSLT14 and Gigaword-10K, diffusion language models (RDM) initialized from a pretrained MLM model considerably outperform the randomly initialized one. This suggests the benefit of self-supervised learning with large-scale data for diffusion language models. Moreover, experimental results show minor improvement on WMT14 (4.5M pairs), a relatively more obvious gain on IWSLT14 (160K pairs),

---

[7]As shown in Tab. 1, diffusion (RDM) slightly underperforms AR with encoder-decoder architectures but largely outperforms in the decoder-only setting, on IWSLT14 translation. A notable difference between the models in these two settings lies in the receptive field. Diffusion always has a global receptive field on the conditioning input, whereas AR can only perceive the condition with unidirectional attention if not equipped with an encoder. This supports our motivation to build diffusion language models for their favorable global receptive field.

[8]The intermediate steps demonstrate that the models generate three clauses simultaneously, implying a global perception that plans the generation of the whole sequence. We consider this benefits the model on more complex generation tasks, which we discuss in §A.2.

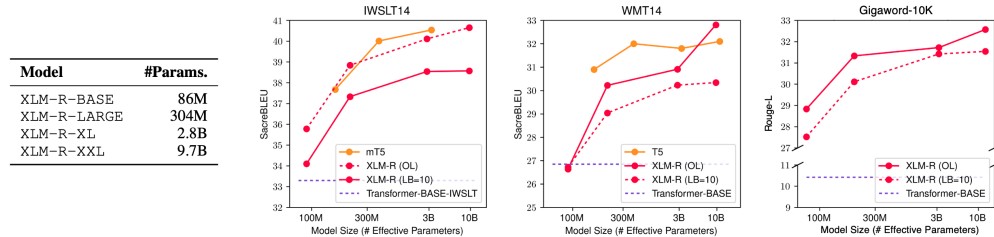

| Model | #Params. |
|---|---|
| XLM-R-BASE | 86M |
| XLM-R-LARGE | 304M |
| XLM-R-XL | 2.8B |
| XLM-R-XXL | 9.7B |

Figure 3: Scaling curves of task-specific finetuning on IWSLT14, WMT14 and Gigaword-10K. We obtain results of mT5 (Xue et al., 2020) on IWSLT14 by ourselves. The results of T5 on WMT14 are from Raffel et al. (2020). "OL": results obtained with oracle target lengths. "LB=10": length prediction results with 10 length beams. "#Params.": Number of effective parameters (*i.e.*, non-embedding parameters).

and a significant performance boost on Gigaword-10K (10K pairs). This demonstrates that the benefit of pretraining is more obvious if the training set of the downstream task is smaller, indicating the effect of pretraining in scaling data.

### 4.3 SCALING UP THE SIZES OF DIFFUSION LMS BOOST DOWNSTREAM TASKS

We now move on to investigate the scalability with respect to model sizes. We finetune XLM-R models of different scales (Conneau et al., 2019; Goyal et al., 2021), whose numbers of effective parameters (*i.e.*, number of non-embedding parameters) range from <100M to 10B. Notably, when the model scales up to 10B, it shows impressive performance that surpasses base-sized models by a remarkable margin (Tab. 1).

Fig. 3 shows the scaling curve of model performance with respect to model sizes. It demonstrates that the performance of the finetuned diffusion models substantially increases as the model size increases. This shows the scaling law of diffusion language models in terms of model size. In addition, we also include the performance of (m)T5 (Raffel et al., 2020; Xue et al., 2020) at similar scales as references to intuitively understand how scalable our diffusion language models are. Note that the performance of different models is intricately affected by not only the model size but also numerous factors including model designs, pretraining budget, pretraining objectives, as well as pretraining data (Shazeer, 2020; Raffel et al., 2020; Tay et al., 2022; Scao et al., 2022; Hoffmann et al., 2022). In Fig. 3, although we see a performance gap between the finetuned (m)T5 and XLM-R models at similar scales, the discrepancy is minor and does not seem amplified as models scale up. Therefore, while there is still ample room for improving large-scale pretrained diffusion language models, we believe that the path of scaling up these models holds great promise.

### 4.4 INSTRUCTION-FINETUNING HELPS GENERALIZE TO UNSEEN TASKS

A fascinating property that motivates scaling language models up is that large language models can follow instructions and show impressive few-shot or even zero-shot performance (Brown et al., 2020; Wei et al., 2021). We now investigate whether diffusion models can also exhibit zero-shot and few-shot performance when being scaled up.

#### 4.4.1 INSTRUCTION FINETUNING ELICITS SCALABLE ZERO-SHOT PERFORMANCE

**Strict zero-shot evaluation on IWSLT14 DE→EN.** We first conduct a strict zero-shot evaluation to study if diffusion language models can acquire zero-shot capabilities through instruction finetuning. Specifically, we evaluate on IWSLT14 DE→EN translation task, for which we instruction-finetune diffusion language models on Flan 2021 Collection (Wei et al., 2021) with all German data removed to ensure that the DE→EN translation becomes a strictly unseen task. As shown in Tab. 2, the instruction-tuned diffusion language models demonstrate scalable zero-shot performance even without finetuning with German data, signifying that large diffusion LMs are able to follow natural language instructions.

Table 2: **Zero-shot** SacreBLEU of instruction-finetuned diffusion language models on IWSLT14 DE→EN translation. For Flan 2021, we explicitly remove all German data for strict evaluation. Results are obtained with oracle length.

| Architecture | Strict Flan'21 | Flan'22 |
|---|---|---|
| *instruction-tuned diffusion:* | | |
| XLM-R-BASE (85M) | 7.15 | 21.26 |
| XLM-R-LARGE (304M) | 22.52 | 25.24 |
| XLM-R-XL (2.8B) | 27.27 | 28.13 |
| XLM-R-XXL (9.7B) | 28.74 | 29.59 |
| ref: *supervised* AR on 160k DE→EN data: 33.30 | | |

**Extensive zero-shot evaluation with large-scale instruction tuning.** We then follow the recommended settings and conduct larger-scale instructions tuning on the full Flan 2022 Collection (Longpre et al., 2023) and run extensive evaluations[9]. Following Chung et al. (2022), we named our instruction-tuned models on Flan 2022 Collection as `Flan-XLM-R`. The results in Fig. 4 suggest that the `Flan-XLM-R` models are indeed general-purpose zero-shot learners, and their zero-shot performance substantially improves as the model scales.

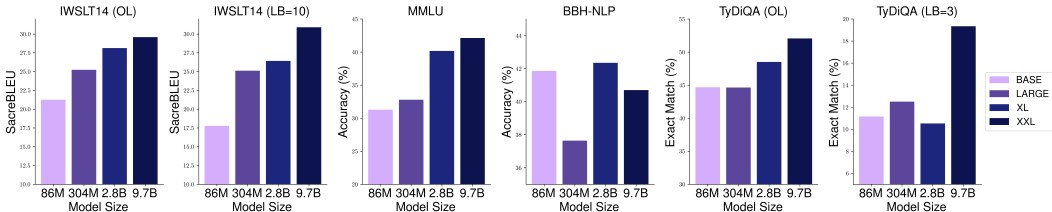

Figure 4: Zero-shot performance of `Flan-XLM-R` models. OL means the results are obtained with oracle length, while LB means the number of length beams to sample the target with length prediction. The model sizes refer to the number of non-embedding parameters.

In particular, we highlight the results on IWSLT14. The largest model, `Flan-XLM-R-XXL` even achieves a 30.90 zero-shot ScareBLEU score, only 2.4 below the performance of widely adopted supervised transformer baselines (33.30 as shown in Tab. 2). This indicates the Flan-XLM-R models produce a very good language generation quality.

### 4.4.2 DIFFUSION LANGUAGE MODELS CAN DO IN-CONTEXT LEARNING

We also evaluate the in-context ability of the large diffusion language models. Limited by the maximum length of 512 `XLM-R` supports, our few-shot evaluation only involves at most 2 demonstrations except for TyDiQA, on which we follow Chung et al. (2022) and evaluate 1-shot performance.

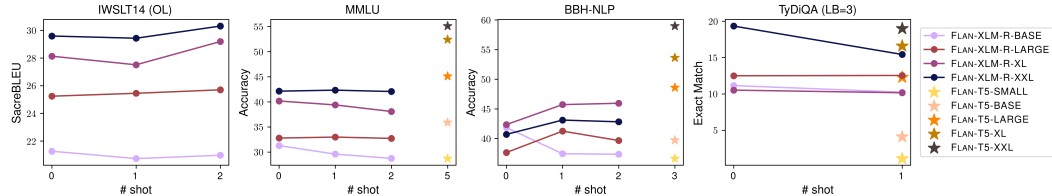

Figure 5: Few-shot performance of `Flan-XLM-R` and `Flan-T5` models. "OL" means the results are obtained with oracle length, while "LB" means the number of length beams to sample the target with length prediction. The model sizes refer to the number of non-embedding parameters.

As shown in Fig. 5, we demonstrate that diffusion language models also obtain the ability to do in-context learning for few-shot settings. We find that the gap between instruction-tuned models' zero-shot performance and in-context few-shot performance is small, which is consistent with similar findings in autoregressive language models (Chung et al., 2022; Longpre et al., 2023; Fu et al., 2023).

### 4.5 EXPLORING REASONING ABILITIES OF DIFFUSION LANGUAGE MODELS

We are also interested in exploring the reasoning abilities of our diffusion language models as it is a crucial emergent ability that distinguishes large language models from the small ones (Wei et al., 2022a; Fu et al., 2023). We highlight our key findings here, and include detailed discussion in §A.

---

[9]We continue to evaluate on the **IWSLT14** dataset. Besides, we also evaluate several datasets used in Chung et al. (2022). In detail, **MMLU** (Hendrycks et al., 2020) includes multiple-choice exam questions from 57 tasks covering elementary mathematics, US history, computer science, law, and more. **BBH-NLP** (Suzgun et al., 2022) covers 12 challenging multiple-choice tasks in BigBench (Srivastava et al., 2022) where language models still underperform the average human-rater. **TyDiQA** (Clark et al., 2020) is an open-book question-answering benchmark across 8 typologically diverse languages

As shown in Fig. 6, we find that even `Flan-XLM-R-XXL` **fails** to emerge non-trivial reasoning performance on GSM8K (Cobbe et al., 2021), a benchmark dataset for mathematical reasoning, and its German translated version in MGSM (Shi et al., 2022). As such, we further conduct in-depth qualitative analysis to gain a fine-grained understanding of the reasoning ability of our models.

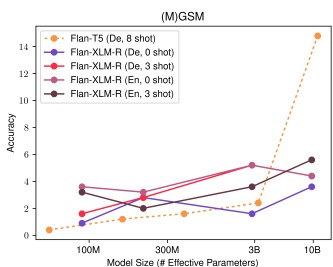

Figure 6: Evaluation on multi-step reasoning on GSM datasets.

**Diffusion LMs could generate content in causal orders.** Non-autoregressive language models typically face the challenge of modeling complex dependencies between target tokens. For reasoning, in particular, models require previously generated tokens (*i.e.*, premises) to improve the generation accuracy of later tokens (*i.e.*, conclusions). Formally, this requires the model to generate tokens conforming to a topological sort on a causal graph (Pearl, 1998). Encouragingly, we find that the generation order of our diffusion language models (1) satisfies this requirement even without specific tuning; and (2) shows a topological sort different from autoregressive models, indicating ability or potential in backtracing and planning. Please refer to §A.2.1 for a concrete example.

**Mitigating the limitation of foundation models probably unlocks the reasoning ability of diffusion language models.** For one thing, we find the accuracy of our models on GSM8K rockets from 5.6% to 12.8% after tuning on chain-of-thought data of GSM8K distilled from `code-davinci-002`, provided by Fu et al. (2023), who show this strategy effective in specializing small models to reason in particular task. Therefore, we suggest that our diffusion language models are able to reason while their generic reasoning ability is limited by the model sizes. Additionally, the pretraining recipe of `XLM-R` differs from current best practices, for which it is poor in some essential abilities like doing calculations. Through comparison (§A.2.2), we find a more up-to-date recipe could benefit the arithmetic abilities of the models.

In summary, we expect more research on the pretraining of diffusion language models to mitigate the limitation of foundation models, unlocking their potential in complex reasoning abilities. We leave this for future exploration.

## 5 DISCUSSIONS, LIMITATIONS, AND FUTURE WORK

In this work, we pioneer studying the scalability of diffusion language models to catch up with recent advances in large language models and facilitate the exploration of their potential. Our investigation comprehensively covers scaling on the data, model sizes, and tasks. Experimental results verify that (1) diffusion language models benefit from large-scale pretraining; (2) their performance improves as the sizes scale up; and (3) they exhibit zero-shot and few-shot capabilities in extensive tasks. While these findings show the promise of large diffusion language models, admittedly, the models are still weak in some advanced abilities like reasoning. Nevertheless, we qualitatively showcase that diffusion language models can generate content in causal orders, showing positive prospects of advanced abilities for future research.

Limitedly, we only build diffusion language models by tuning existing large masked language models instead of pretraining from scratch. However, there exist large discrepancies in architecture and data engineering between our foundation models, `XLM-R` (Conneau et al., 2019), which were built years ago, and the state-of-the-art large language models like `LLaMA` (Touvron et al., 2023a;b). This impedes us from approaching the ultimate capability of current diffusion language models. Evident limitations include the fairly short maximum length (*i.e.*, 512) and unsatisfying arithmetic ability. Additionally, the difference in masking ratios also questions whether diffusive adaptation is enough to fill the gap between BERT-like pretraining and diffusion models. Therefore, there remains a great need to investigate pretraining for diffusion language models in future research.

Overall, our study confirms the scalability of diffusion language models and leads to future research on the continual exploitation of large diffusion language models. Compared with autoregressive models, diffusion language models are probabilistically more expressive (Gong et al., 2022) and cover a more extensive set of languages (Lin et al., 2021). Practically, they enjoy a global receptive field and generate via non-autoregressive iterative refinement, potentially bringing advanced capabilities, such as supporting drafting-then-revising and backtracking manners in nature. We hope that our findings will facilitate the success of diffusion models in broader domains and also encourage a compelling alternative to autoregressive large language models, which might push forward the boundary of techniques to pursue more advanced machine intelligence.

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

## A    Exploring Reasoning Abilities of Diffusion Language Models

We are interested in whether diffusion language models can solve tasks by multi-step reasoning as this is a crucial emergent ability that marks the success of large language models and distinguishes them from the small ones (Wei et al., 2022a; Fu et al., 2023). Chung et al. (2022) shows that with a certain amount of chain-of-thought instruction tuning data (included in the Flan 2022 Collection), it is possible for 10B models to emerge reasoning ability to a certain extent. This motivates us to investigate the reasoning ability of our diffusion language models as our largest XLM-R does scale up to this size.

### A.1    Quantitative Evaluation

We first evaluate our instruction-tuned models on common reasoning benchmarks to gain a coarse understanding of the reasoning abilities of diffusion language models.

**Evaluation setup.** We studied reasoning abilities on **GSM8K** (Cobbe et al., 2021) which contains diverse grade school math word problems, and its translated version **MGSM** (Shi et al., 2022). Previous studies (Wei et al., 2022b) have shown solving them requires multi-step reasoning which is a typical emergent ability that only exists in models large enough (Wei et al., 2022a). Even the largest Flan-T5 (*i.e.*, Flan-T5-11B) can only show plausible performance on some of the MGSM subsets of high-resource languages such as German, Spanish, and French (Chung et al., 2022). Therefore, for MGSM, we skip the low-resource languages and only evaluate the German subset. For implementation details, we use the chain-of-thought prompting from Shi et al. (2022). We set 60 as the target length in zero-shot evaluation, and use the length of the longest demonstrations for few-shot evaluation.

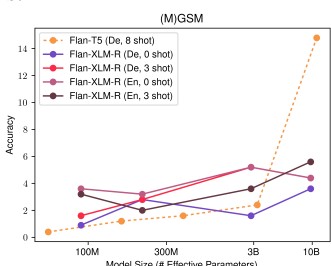

Figure 7: Evaluation on math word problems in (M)GSM (Cobbe et al., 2021; Shi et al., 2022) with step-by-step answer promptings.

**Diffusion language models still fall short of complex reasoning tasks.** As shown in Fig. 7, our instruction-tuned XLM-R fails to emerge with positive reasoning performance on all the evaluated settings. Considering that solving math word problems requires multi-step reasoning, their correctness may only emerge as the quality of all intermediate steps improves to a certain extent. To this end, we next focus on the qualitative analysis of the reasoning steps to gain a fine-grained understanding of reasoning ability of our models.

### A.2    Qualitative Analysis

A typical challenge in non-autoregressive language modeling arises from the complex dependencies between target tokens (Zhou et al., 2020; Gu & Tan, 2022). This, we suggest, also explains why our diffusion language models excel at traditional generation tasks but struggle with reasoning. Specifically, tasks like machine translation and text summarization have strong conditioning, wherein most of the target tokens can be predicted based on the given conditions. The intermediate steps mainly function to resolve conflicts between different possible results (*a.k.a.*, multimodality problem (Gu et al., 2018)). However, in reasoning tasks, a model need to generate intermediate reasoning steps to approach the final answers (*a.k.a.*, "let's think step by step"). In this case, the model more heavily relies on the intermediate results generated by itself to make predictions. This leads to constraints on the generation order when performing reasoning tasks.

We now further elaborate on this from the perspective of causal graphs for reasoning tasks.

### A.2.1    Understanding target dependencies with causal graphs.

We consider illustrating the semantic dependencies in reasoning tasks with causal graphs (Pearl, 1998), directed acyclic graphs whose edges point from causes to their effects. Fig. 8(a) depicts the causal graph for the exemplary problem and its solution shown in Fig. 8(b). We argue that, in order to solve the task with reasoning, language models must generate tokens in an order that conforms to a *topological sort* of the causal graph. Specifically, it means the following requirements for the generation order: (1) the final results should come after the last intermediate result; (2) the intermediate results should come after listing the corresponding equation; (3) to correctly list an

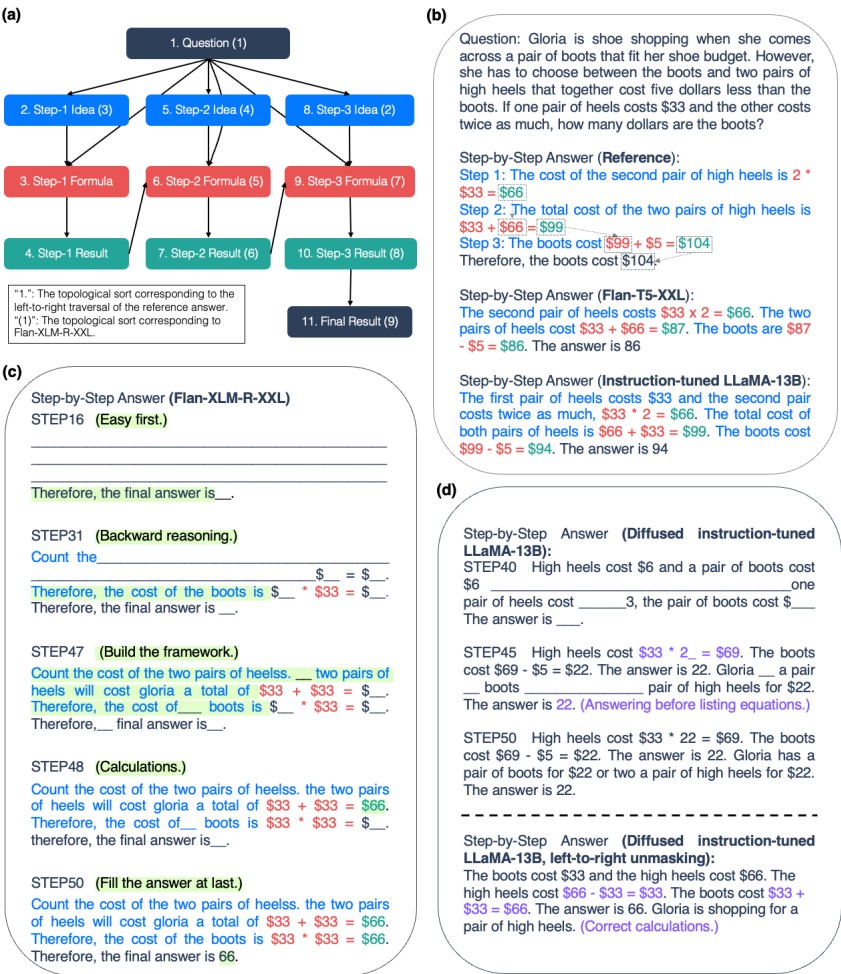

Figure 8: Qualitative investigation into the reasoning abilities of diffusion language models. **(a)** A causal graph (Pearl, 1998) that represents the dependencies between the reasoning steps. **(b)** An example question, its reference answer, and answers from autoregressive models. **(c)** The answer from `Flan-XLM-R-XXL` and its generation process. **(d)** Answers from a diffusive instruction-tuned `LLaMA-13B`, sampling with diffusive unmasking or left-to-right unmasking strategy, where the former fails to generate in an order conforming to one of the topological sorts in (a).

equation, models need to have the idea for this equation, copying calculation results from previous steps or numbers provided by the question; and (4) before these, models need to propose the idea for each step first.

**Diffusion language models can figure out feasible topological sorts on the causal graph.** A follow-up question is whether the generation process of autoregressive models and our diffusion language models conform to possible topological sorts. In fact, one feasible topological sort is exactly the left-to-right traversal on the chain-of-thought text and is implicitly provided to autoregressive models during training. Diffusion language models, on the other hand, learn without a fixed generation order due to random masking. To see if they can figure out a feasible generation order, we demonstrate its generation process of solving the exemplary question in Fig. 8(c). Encouragingly, despite incorrect final answers, the generative process does conform to a topological sort of the causal graph in Fig. 8(a). The model generates the ideas first, then writes the formulas, and finally calculates the answers[10]. This implies that diffusion language models learn to figure out feasible topological sorts, namely a structure reasoning ability.

---

[10]We also observe similar behaviors in other examples. For clarity, we focus on one of them in this paper and include others in our open-sourced repository.

**Diffusion language models reason with a flexible mind.** Notably, diffusion language models are able to explore different topological sorts different from autoregressive models thanks to less constrained generative orders. We highlight some of the interesting patterns resulting from this.

- **Easy first.** Fig. 8(c) shows that the model fills up the fixed pattern (*i.e.*, "the final answer is") at first, showing a quite smart easy-to-hard generation behavior.
- **Planning ahead.** In Fig. 8(c), the model constructs the framework for the solution before diving into arithmetic. Actually, we have seen similar behavior in Fig. 2 where the model generates three clauses simultaneously. Both cases demonstrate the models' global perception which helps plan the generation of the whole sequence.
- **Forward and backward reasoning.** During the reasoning process in Fig. 2, on STEP 31, the model begins the solution with the idea for the last reasoning step. This shows backward reasoning behavior, a very common human behavior that is especially helpful for challenging reasoning activities such as finding mathematical proofs (Kazemi et al., 2022).
- **Backtracing.** The backward transition of diffusion models (Eqn. 4) formally supports backtracing by remasking tokens. In Fig. 8(c), STEP 47 erases a "the" token. This ability helps avoid accumulating errors in predicted tokens (Arora et al., 2022).

These observations signify the potential to elicit diffusion language models' reasoning abilities beyond chain-of-thought whose limitations have been demonstrated in Yao et al. (2023). We believe this can encourage further research on the reasoning ability of diffusion language models.

### A.2.2 UNDERSTANDING THE IMPACT OF FOUNDATION MODELS.

Although showing a flexible generation manner, admittedly, our diffusion language models rarely predict the correct answers. One superficial hypothesis is that `XLM-R` models have limited capability. We elaborate on this as follows.

- **Limitations on model sizes.** The largest `XLM-R` model only scales up to around 10B, a borderline to demonstrate plausible reasoning performance for previous instruction-tuned autoregressive models (Chung et al., 2022). It is likely that our diffusion language models follow a different scaling law (Kaplan et al., 2020; Hoffmann et al., 2022) from that of autoregressive models, and we need to further enlarge the model to elicit its reasoning ability. Alternatively, Fu et al. (2023); Magister et al. (2022) succeed in specializing smaller language models to solve certain reasoning tasks (*e.g.*, math reasoning) (Fu et al., 2023; Magister et al., 2022) by distilling from large and capable models. We consider it promising to explore similar attempts for diffusion language models. For one thing, we can leverage more capable models (*e.g.*, GPT4) to obtain specialized reasoning data for distillation[11]. On the other hand, prestigious models can serve by discovering the causal graph in the reasoning data (Zhang et al., 2023; Kıcıman et al., 2023), with which we can involve *process supervision* (Uesato et al., 2022; Lightman et al., 2023) to facilitate the learning of generation order.
- **Limitations on training recipe.** `XLM-R` are pretrained on a recipe largely different from the best practices of the current state-of-the-art large language models. For instance, the data include little to no code or scientific literature, which are hypothesized to be crucial for reasoning ability (Taylor et al., 2022; Lewkowycz et al., 2022). As an attempt, we tried converting a more recent and capable autoregressive model `LLaMA` (Touvron et al., 2023a) into diffusion language models to handle these limitations. Specifically, we initialized the model with a 13B `LLaMA` and instruction-tuned it with the diffused training objective (Eq. 5). With `LLaMA`, we find the instruction-tuned model does showcase better arithmetic ability (Fig. 8(d), bottom). However, we find it fails to generate in an order that conforms to a topological sort of the corresponding causal graph in Fig. 8(a). For instance, in STEP 45 of Fig. 8(d)'s upper part, the model generates calculation results before completing the formula and also the final result before the intermediate results. This implies two critical influences of the foundation models. First, a more up-to-date pretraining recipe is helpful for arithmetic abilities. Second, the limitation in the training recipe cannot be simply bypassed by applying diffusive adaptation to competent autoregressive models due to the discrepancy in training objectives, and diffusion language models' potential to perform structure reasoning is probably a product of masked language modeling pretraining. These two implications naturally

---

[11]As a verification, we tried finetuning `Flan-XLM-R-XXL` on GSM8K chain-of-thought training data distilled by Fu et al. (2023) from `code-davinci-002`. After this specialization, our model's zero-shot and 3-shot performance on GSM8K rocket from 4.4% and 5.6% to 10.0% and 12.8%, respectively.

lead to the idea of improving diffusion language models' reasoning ability by pretraining with a more up-to-date recipe (*e.g.*, the RedPajama[12]).

We leave these improvements on foundation models as future work and encourage further research on them.

## B RELATED WORK

**Language Modeling** aims to learn a probabilistic model to describe sequence data $p(\boldsymbol{x}^{[1:N]})$ of interest (Shannon, 1951; Jurafsky & Martin, 2009). The dominant paradigm, autoregressive language models, decomposes the joint distribution over the tokens of a sequence into conditionals with the chain rule $p(\boldsymbol{x}^{[1:N]}) = \prod_{i=1}^{N} p(x^{[i]}|\boldsymbol{x}^{[1:i-1]})$ and generates tokens by ancestral sampling from left to right (Bengio et al., 2000; Sutskever et al., 2014; Vaswani et al., 2017). Recent advances propose non-autoregressive language models as an alternative (Gu et al., 2018). They circumvent the constraint of a predefined generation order (Qian et al., 2022; Huang et al., 2023) and show competitive or superior performance compared to their autoregressive counterpart in various domains including languages (Qian et al., 2021; Huang et al., 2022; Qian et al., 2022; Huang et al., 2023; Zheng et al., 2023a), speeches (Kim et al., 2021), proteins (Zheng et al., 2023b), and molecules (Hoogeboom et al., 2022). Among various non-autoregressive language models, diffusion language models (Li et al., 2022; Gong et al., 2022; Zheng et al., 2023a) have recently arisen as a promising and theoretically grounded paradigm.

**Large Language Models.** Pretraining language models on massive unlabeled data dramatically boost their performance on downstream tasks (Mikolov et al., 2013; Peters et al., 2018; Radford et al., 2018; Devlin et al., 2018). As data volume and model sizes scale up, the training loss language models reach predictably decreases (Kaplan et al., 2020; Hoffmann et al., 2022; Muennighoff et al., 2023), and performance improves across tasks even without specific finetuning (Radford et al., 2019). A milestone to this end is GPT3 (Brown et al., 2020), which scales the models up to 175B parameters and proposes in-context learning to elicit language models' ability to solve specific tasks with only a few demonstrations. Wei et al. (2021); Sanh et al. (2022); Ouyang et al. (2022) further introduce instruction tuning, finetuning pretrained language models on collections of tasks described via instructions, to improve their zero-shot performance in unseen tasks. More impressively, sufficiently large language models emerge with advanced abilities such as multi-step reasoning (Kojima et al., 2022; Wei et al., 2022a;b), differentiating them from small models (Fu et al., 2023). Empowered by large language models, helpful applications such as conversational AI systems[13] and autonomous agents[14] have drawn great attention. Although the most capable models for the time being remain close-sourced, open-sourced efforts (Zeng et al., 2022; Touvron et al., 2023a;b; Taori et al., 2023; Chiang et al., 2023; Sun & Qiu, 2023) have largely enhanced the public accessibility of powerful large language models. While most existing works are based on autoregressive language models, our study investigates scaling diffusion language models, a kind of non-autoregressive language models.

**Diffusion Language Models** are language models based on diffusion models (Sohl-Dickstein et al., 2015), a type of generative model that samples data via iterative denoising from noise, which can be categorized into continuous (Ho et al., 2020; Song et al., 2020) and discrete (Hoogeboom et al., 2021; Austin et al., 2021) ones according to the distribution they model. Despite huge success in vision (Dhariwal & Nichol, 2021; Rombach et al., 2021; Ho et al., 2022), continuous diffusion models for languages that operate on continuous surrogates of discrete tokens (Li et al., 2022; Gong et al., 2022; Han et al., 2022; Dieleman et al., 2022; Yuan et al., 2022; Gao et al., 2022; Ye et al., 2023; Chen et al., 2023; Wu et al., 2023) struggle to overcome the pitfall of discreteness (Ye et al., 2023) and still lag behind autoregressive language models. On the other hand, discrete diffusion models have limited progress in large-scale applications but they naturally fit the data type of languages (*i.e.*, sequences of discrete tokens). Recent advancement by Zheng et al. (2023a) successfully improves them and achieves comparable performance with autoregressive models on typical language generation benchmarks like machine translation. Moreover, He et al. (2023); Zheng et al. (2023b) show the close relationship between discrete diffusion models and masked language models, a widely adopted pretraining paradigm in NLP (Devlin et al., 2018; Liu et al., 2019), implying the possibility to build large discrete diffusion language models. Motivated by these findings, in this work, we

---

[12] https://github.com/togethercomputer/RedPajama-Data
[13] https://chat.openai.com/
[14] https://github.com/Significant-Gravitas/Auto-GPT

investigate the scalability of diffusion language models to explore their potential further. There are also recent attempts to pretrain continuous diffusion language models (). In comparison, our work builds on discrete diffusion models and leverages their connection to publicly accessible masked language models, for which we can explore larger models and their general-purpose ability such as instruction following (Lin et al., 2022; Balagansky & Gavrilov, 2023; Gulrajani & Hashimoto, 2023). The most relevant work to ours is Han et al. (2023) which builds a 13B chat model with continuous simplex-based diffusion language models. In contrast, our work focuses on discrete diffusion language models and their general abilities on diverse tasks.

## C    REPARAMETERIZAED DISCRETE DIFFUSION MODELS (RDM)

Zheng et al. (2023a) shows that the backward transition of discrete diffusion models $q(\boldsymbol{x}_{t-1}|\boldsymbol{x}_t, \boldsymbol{x}_0)$ can be rewritten as

$$
q(\boldsymbol{x}_{t-1}|\boldsymbol{x}_t, \boldsymbol{x}_0) = \begin{cases} \texttt{Cat}\left(\boldsymbol{x}_{t-1}; \boldsymbol{p} = \lambda_{t-1}^{(1)}\boldsymbol{x}_t + (1 - \lambda_{t-1}^{(1)})\boldsymbol{q}_{\text{noise}}\right), & \text{if } \boldsymbol{x}_t = \boldsymbol{x}_0 \\ \texttt{Cat}\left(\boldsymbol{x}_{t-1}; \boldsymbol{p} = \lambda_{t-1}^{(2)}\boldsymbol{x}_t + (1 - \lambda_{t-1}^{(2)})\boldsymbol{q}_{\text{noise}}(\boldsymbol{x}_t)\right), & \text{if } \boldsymbol{x}_t \neq \boldsymbol{x}_0 \end{cases}, \quad (4)
$$

where $\boldsymbol{q}_{\text{noise}}(\boldsymbol{x}_t) = \beta_t \boldsymbol{x}_t + (1 - \beta_t)\boldsymbol{q}_{\text{noise}}$, and both $\lambda_{t-1}^{(1)}$ and $\lambda_{t-1}^{(2)}$ are constants relating to $\beta_t$ and $\beta_{t-1}$. This reformulation interprets the backward transition as a mixture distribution. Sampling from it is equivalent to first sampling from a Bernoulli distribution and then the corresponding component distribution, *i.e.*,

$$
v_{t-1}^{(1)} \sim \text{Bernoulli}\left(\lambda_{t-1}^{(1)}\right), \qquad \boldsymbol{u}_t^{(1)} \sim \texttt{Cat}\left(\boldsymbol{u}; \boldsymbol{p} = \boldsymbol{q}_{\text{noise}}\right),
$$

$$
v_{t-1}^{(2)} \sim \text{Bernoulli}\left(\lambda_{t-1}^{(2)}\right), \quad \boldsymbol{u}_t^{(2)} \sim \texttt{Cat}\left(\boldsymbol{u}; \boldsymbol{p} = \boldsymbol{q}_{\text{noise}}(\boldsymbol{x}_t)\right),
$$

$$
\boldsymbol{x}_{t-1} = \begin{cases} v_{t-1}^{(1)}\boldsymbol{x}_t + \left(1 - v_{t-1}^{(1)}\right)\boldsymbol{u}_t^{(1)}, & \text{if } \boldsymbol{x}_t = \boldsymbol{x}_0 \\ v_{t-1}^{(2)}\boldsymbol{x}_t + \left(1 - v_{t-1}^{(2)}\right)\boldsymbol{u}_t^{(2)}, & \text{if } \boldsymbol{x}_t \neq \boldsymbol{x}_0 \end{cases}.
$$

This reparameterizes the transitions in Eq. (1) into $q(\boldsymbol{x}_t, \boldsymbol{v}_{t-1}|\boldsymbol{x}_t, \boldsymbol{x}_0)$ and $p_{\boldsymbol{\theta}}(\boldsymbol{x}_{t-1}, \boldsymbol{v}_{t-1}|\boldsymbol{x}_t)$. With this reparameterization, the training objective of diffusion models (*i.e.*, the variational bound of negative log-likelihood) becomes

$$
-\mathbb{E}_q(\boldsymbol{x}_{1:T}, \boldsymbol{v}_{1:T}|\boldsymbol{x}_0)\left[\log \frac{p_{\boldsymbol{\theta}}(\boldsymbol{x}_0, \boldsymbol{x}_{1:T}, \boldsymbol{v}_{1:T})}{q(\boldsymbol{x}_{1:T}, \boldsymbol{v}_{1:T}|\boldsymbol{x}_0)}\right] = L_1 + \sum_{t=2}^{T} L_t + \text{const.},
$$

where $L_1 = -\mathbb{E}_{q(\boldsymbol{x}_1|\boldsymbol{x}_0)}\left[\log p_{\boldsymbol{\theta}}(\boldsymbol{x}_0|\boldsymbol{x}_1)\right]$ and Zheng et al. (2023a) shows that $L_t$ can be simplified into

$$
L_t = \mathbb{E}\left[-\lambda_{t-1}^{(2)}\left(1 - \mathbb{1}(\boldsymbol{x}_t = \boldsymbol{x}_0)\right)\log p_{\boldsymbol{\theta}}(\boldsymbol{x}_0|\boldsymbol{x}_t)\right], \quad (5)
$$

where $\mathbb{1}(\cdot)$ is indicator function. This is exactly a weighted cross-entropy loss on the perturbed data, *a.k.a.* the masked language modeling objective (Devlin et al., 2018). Notably, training with different noise schedules only differs in the weighting of the objective. During sampling, RDM leverages this observation and proposes to employ a discriminative approach. Specifically, it denoises a token only when it receives a top-$k$ score (log-probability) from the network where $k$ in each step is determined by a denoising schedule.

## D    IMPLEMENTATION DETAILS

### D.1    MODEL

Throughout this work, we mainly follow Zheng et al. (2023a) to train and sample from our diffusion language models. Specifically, we set $\lambda_{t-1}^{(2)} = 1 - \frac{t-1}{T}$ in the training objective (Eqn. 5) where $t$ is the current timestep and $T$ is the number of total timesteps which is 50 in our experiments. Additionally, we apply label smoothing with a factor of 0.1 when we train a model without pretraining. During sampling, we also follow Ghazvininejad et al. (2019); Savinov et al. (2021); Zheng et al. (2023b) and denoise tokens with high scores in each step instead of naively sampling from the Bernoulli

distributions. We use the same cosine schedule as in Zheng et al. (2023a) to decide the number of denoised tokens in each step $k = \lfloor N \cdot \cos \frac{\pi t}{2T} \rfloor$, where $N$ is the sequence length. For full details, we refer readers to the pseudocode in the original paper (Zheng et al., 2023a, Algorithm 2). Besides, we follow the time agnostic design in He et al. (2023) that does not introduce any extra parameters to differentiate different timesteps. For length prediction, we feed model outputs into a one-layer transformer, apply mean pooling to the features and feed the pooled feature into an MLP classifier head. For task-specific finetuning, we remove both input and output embeddings of the tokens that do not appear in the training set.

## D.2 DATA

For IWSLT14 and WMT14 machine translation tasks, we download and preprocess data following the example scripts provided by `Fairseq`[15], and we use SacreBleu (Post, 2018) for evaluation[16]. And we download Gigaword-10K data from the repository of LGEB[17]. For (M)GSM, we follow the instruction[18] in the official repository of Shi et al. (2022) to process the data and prompts. Besides, we obtain the preprocessed Flan 2021[19], Flan 2022[20], MMLU[21], BBH[22] and TydiQA[23] from shared datasets on HuggingFace[24]. During training with Flan 2022, we follow the recommended ratios in Chung et al. (2022) to sample training data from different subsets. We follow Chung et al. (2022) to report the MMLU performance on the validation set and adopt the GoldP setting for TyDiQA as in Chowdhery et al. (2022); Chung et al. (2022). On the few-shot settings, we randomly select demonstrations. We will also release our code and data for better reproducibility.

## D.3 TRAINING DETAILS

We use Adam optimizer (Kingma & Ba, 2015) throughout our study. The dropout rate is consistent with the original configuration of the models which is 0.1. For task-specific tuning, we use 8 Nvidia A100 GPUs. For instruction tuning, we use 8 Nvidia V100 GPUs for BASE and LARGE-sized models, 32 for XL, and 64 for XXL. The overall batch size and other detailed hyperparameters for the two settings are in Tab. 3 and Tab. 4, respectively.

Table 3: The training hyperparameters for task-specific finetuning.

| Dataset | Pretrained model | Batch size (#. tokens) | Learning rate | #. training steps |
|---|---|---|---|---|
| IWSLT14 DE→EN | XLM-R-BASE | 32K | 5e-5 | 150,000 |
| | XLM-R-LARGE | 32K | 5e-5 | 150,000 |
| | XLM-R-XL | 32K | 5e-5 | 100,000 |
| | XLM-R-XXL | 32K | 5e-5 | 30,000 |
| WMT14 EN→DE | XLM-R-BASE | 128K | 5e-5 | 300,000 |
| | XLM-R-LARGE | 128K | 5e-5 | 300,000 |
| | XLM-R-XL | 128K | 5e-5 | 150,000 |
| | XLM-R-XXL | 128K | 5e-5 | 100,000 |
| Gigaword-10K | XLM-R-BASE | 16K | 5e-5 | 30,000 |
| | XLM-R-LARGE | 16K | 5e-5 | 10,000 |
| | XLM-R-XL | 16K | 5e-5 | 5,000 |
| | XLM-R-XXL | 16K | 5e-5 | 1,000 |

---

[15]https://github.com/facebookresearch/fairseq/tree/main/examples/translation

[16]The signature of sacrebleu for IWSLT14 DE→EN is `nrefs:1|case:mixed|eff:no|tok:13a|smooth:exp|version:2.3.1`, and for WMT14 EN→DE `nrefs:1|case:mixed|eff:no|tok:intl|smooth:exp|version:2.3.1`, respectively.

[17]https://github.com/CLUEbenchmark/LGEB

[18]https://github.com/google-research/url-nlp/tree/main/mgsm

[19]https://huggingface.co/datasets/Muennighoff/flan

[20]https://huggingface.co/datasets/SirNeural/flan_v2

[21]https://huggingface.co/datasets/cais/mmlu

[22]https://huggingface.co/datasets/lukaemon/bbh

[23]https://huggingface.co/datasets/khalidalt/tydiqa-goldp

[24]https://huggingface.co/datasets

Table 4: The training hyperparameters for instruction finetuning.

| Training data | Pretrained model | Batch size (#. sequence) | Learning rate | #. training steps |
|---|---|---|---|---|
| Flan 2021 | XLM-R-BASE | 512 | 5e-5 | 5,000 |
| | XLM-R-LARGE | 512 | 5e-5 | 5,000 |
| | XLM-R-XL | 512 | 5e-5 | 3,000 |
| | XLM-R-XXL | 256 | 5e-5 | 1,000 |
| Flan 2022 | XLM-R-BASE | 512 | 1e-5 | 70,000 |
| | XLM-R-LARGE | 512 | 1e-5 | 30,000 |
| | XLM-R-XL | 1024 | 1e-5 | 17,000 |
| | XLM-R-XXL | 2048 | 1e-5 | 4,000 |

# E  FULL EXPERIMENTAL RESULTS

The experimental results for task-specific tuning and instruction tuning on Flan 2022 are in Tab. 5 and Tab. 6, respectively.

Table 5: Full experimental results of task-specific finetuning. OL: the results are obtained with oracle length. LB: the size of length beam for length prediction.

| Dataset (Metric) | Setting | XLM-R-BASE | XLM-R-LARGE | XLM-R-XL | XLM-R-XXL |
|---|---|---|---|---|---|
| IWSLT14 DE→EN (SacreBLEU) | OL | 35.78 | 38.84 | 40.11 | 40.65 |
| | LB=10 | 34.10 | 37.33 | 38.54 | 38.57 |
| WMT14 EN→DE (SacreBLEU) | OL | 26.65 | 30.22 | 30.91 | 32.81 |
| | LB=10 | 26.72 | 29.04 | 30.23 | 30.34 |
| Gigaword-10K (Rouge-L) | OL | 28.83 | 31.33 | 31.72 | 32.57 |
| | LB=10 | 27.52 | 30.11 | 31.42 | 31.54 |

Table 6: Full experimental results of instruction tuning on Flan 2022. OL: the results are obtained with oracle length. LB: the size of length beam for length prediction.

| Dataset (Metric) | Setting | XLM-R-BASE | XLM-R-LARGE | XLM-R-XL | XLM-R-XXL |
|---|---|---|---|---|---|
| IWSLT14 DE→EN (SacreBLEU) | 0-shot (OL) | 21.26 | 25.24 | 28.13 | 29.59 |
| | 2-shot (OL) | 20.97 | 25.70 | 29.19 | 30.31 |
| | 0-shot (LB=3) | 17.76 | 25.12 | 26.42 | 30.90 |
| | 2-shot (LB=3) | 15.91 | 23.49 | 27.29 | 31.04 |
| MMLU (Accuracy%) | 0-shot | 31.28 | 32.79 | 40.17 | 42.13 |
| | 2-shot | 28.74 | 32.72 | 38.08 | 42.06 |
| BBH-nlp (Accuracy%) | 0-shot | 41.86 | 37.64 | 42.35 | 40.70 |
| | 2-shot | 37.35 | 39.66 | 45.95 | 42.82 |
| TyDiQA (Exact Match) | 0-shot (OL) | 44.68 | 44.64 | 48.50 | 52.06 |
| | 1-shot (OL) | 44.69 | 48.46 | 49.43 | 51.71 |
| | 0-shot (LB=3) | 11.15 | 12.50 | 10.52 | 19.34 |
| | 1-shot (LB=3) | 10.24 | 12.54 | 10.16 | 15.43 |
| MGSM (DE) (Accuracy%) | 0-shot | 0.9 | 2.8 | 1.6 | 3.6 |
| | 3-shot | 1.6 | 2.8 | 5.2 | 4.4 |
| GSM8K (Accuracy%) | 0-shot | 3.6 | 3.2 | 5.2 | 4.4 |
| | 3-shot | 3.2 | 2.0 | 3.6 | 5.6 |

