# OpenReview forum: "Diffusion Language Models Can Perform Many Tasks with Scaling and Instruction-Finetuning"
_ICLR.cc/2024/Conference — Submitted to ICLR 2024_

### Official Review · Reviewer_xgDh · 2023-10-15

**Soundness:** 2 fair
**Presentation:** 3 good
**Contribution:** 2 fair
**Rating:** 5
**Confidence:** 4

**Summary:**

The paper proposes a method to adapt pretrained masked language models to diffusion language models, using it to adapt MLMs of different sizes to diffusion LMs, and then perform extensive task finetuning and instruction finetuning on the diffusion LMs. Results show that diffusion LMs can perform similarly to autoregressive LMs on tasks and generalize to unseen tasks after instruction finetuning. Diffusion LMs are also shown to manifest in-context learning and reasoning abilities, suggesting that diffusion LMs could have similar capabilities as autoregressive LMs.

**Strengths:**

* This is one of the first papers to investigate the scalability of diffusion LMs, highlighting the potential of diffusion LMs with the increase of model size. The results could potentially draw more attention to and stimulate further research on non-autoregressive LMs.
* The proposed diffusion adaptation technique bridges existing masked LM with diffusion LM using lightweight continual pre-training. It potentially serves as an effective strategy for training diffusion LMs under computation constraints.

* The paper provides a comprehensive evaluation of diffusion LM using multiple supervised, few-shot, and zero-shot tasks, giving an overall complete perspective on the performance of diffusion LMs.

**Weaknesses:**

* Discussion is limited on diffusion adaptation, a core method contribution of the paper. When using an existing MLM checkpoint instead of extensive pre-training, the performance of diffusion LM hinges on the effectiveness of diffusion adaptation. However, the paper does not provide enough performance metrics to validate the effectiveness of diffusion adaptation. For example, how does it compare to training from scratch, and how does the number of adaptation steps affect downstream performance? Without these results, it is unclear if diffusion adaptation is an effective approach to building diffusion LMs.

* While showing diffusion LM can be promising and shows similar capabilities as auto-regressive LM, it is hard to get a concrete understanding of the performance of diffusion LM from the current paper.

  * No effective performance comparison between diffusion LM and other kinds of LMs. While it is convenient to adapt an existing MLM to diffusion LMs and evaluate it, as the authors pointed out, the current available MLM checkpoints are out-of-date, and diffusion adaptation inevitably loses performance compared to training from scratch with diffusion objective. This makes the current result potentially considerably inferior to what diffusion LM can achieve in theory (with a state-of-the-art training recipe and trained from scratch). As a result, it is hard to appreciate the exact capabilities of diffusion LM from the current results. This could also make some discussions ineffective, for instance, it cannot be ruled out that the deficiencies in reasoning are due to insufficient training.

  * Discussion on in-context learning and reasoning abilities is vague/ineffective. From Figure 5, it is hard to observe performance improvement due to in-context learning. For auto-regressive LM, in-context learning can be evaluated on raw pre-trained LM prior to instruction-finetuning. Maybe a similar evaluation on diffusion LMs could better visualize their in-context learning abilities. Also, as the authors pointed out, the limited context length of the original model limits the evaluation of few-shot performance. One possible solution without training from scratch could be using interpolation of position embeddings (e.g. [1]).

    Similarly, there is no concrete conclusion from the discussion on reasoning abilities due to insufficient reasoning performance. One interesting discussion is on the generation order of diffusion LM and its difference from auto-regressive LMs, but a concrete conclusion would probably require a benchmark or a probe dataset for a systematic evaluation, showing the advantage of the generation order employed by diffusion LMs.

  * In my opinion (without any discouragement to the authors), the evaluation of diffusion LMs is probably better done with sufficient computing resources for a head-to-head comparison with auto-regressive LMs under similar training conditions. The results would be much more useful and enable effective discussion on the capabilities of diffusion LMs.

**Questions:**

* How does the performance of diffusion LM compare to other non-autoregressive language models?
* What about the inference computation cost of diffusion LMs, compared to auto-regressive LMs? How does the number of denoising steps affect the performance-computation tradeoff, as it is a crucial parameter in image diffusion models?
* The performance comparison in Table 1 is a bit confusing: 1) are the RDM models trained by the authors? if so, why not evaluate on all three datasets? 2) "The performance of finetuned XLM-R models is competitive or superior to the common encoder-decoder Transformer baseline." may not be so safe to say as the decoder-only models are larger than encoder-decoder models. 3) how is XLM-R (autoregressive) evaluated on IWSLT14, as it is an encoder-only model?
* In Table 2, why not use zero-shot AR as a reference but use supervised AR?
* Do the AR models used in the paper use beam search during decoding?
* Given that diffusion LMs use a different receptive field and generation order than auto-regressive LMs, could there be specific types of tasks or domains where diffusion LMs particularly excel or fall short? That could be a valuable discussion (but probably beyond the scope of the paper).

---

> ### Author Response · Authors · 2023-11-21
>
> Thank you so much for your insightful and constructive suggestions! We would like to address your concerns below. Please have a check and we are happy to address any further feedback!
>
> $$$$
>
> > `Q1:` Discussion is limited on diffusion adaptation, a core method contribution of the paper. The paper does not provide enough performance metrics to validate the effectiveness of diffusion adaptation. For example, how does it compare to training from scratch, and how does the number of adaptation steps affect downstream performance? It is unclear if diffusion adaptation is an effective approach to building diffusion LMs.
>
> Thank you for your question!
> We would like to clarify that our study does not aim to find the best practice for building diffusion language models. Instead, we hope to conduct a pioneering investigation into the scalability of diffusion language models without consuming huge computational resources. Our positive findings support this goal and can stimulate more future efforts on building large diffusion language models, either by pretraining from scratch or diffusive adaption.
>
> $$$$
>
> > `Q2:` No effective performance comparison between diffusion LM and other kinds of LMs.
>
> Conducting a strict comparison across the recipes of different pretrained models can be challenging, especially given the expenses involved and the abundance of technical details. As we respond in Q1, we are not aiming to beat other language models in this study, but to validate the scalability of diffusion language models and encourage further research efforts on this. And our results in Table 1, Figure 3, and Table 2 can support that our models have decent performance compared to models at similar scales, and demonstrate generalist ability that can follow instructions to solve various tasks.
>
> $$$$
>
> > `Q3:` Discussion on in-context learning and reasoning abilities is vague/ineffective. From Figure 5, it is hard to observe performance improvement due to in-context learning.
>
> Observation in Figure 5 is actually consistent with the findings of the FLAN series paper [1,2], which shows that model performance is not sensitive to the number of demonstrations after instruction tuning, and we have mentioned this phenomenon in Section 4.2.2. Note that the most important message here is that instruction tuning helps our Diffusion LMs obtain in-context learning ability, without which we found that a raw pretrained MLM cannot learn from demonstrations.
>
> --
>
> [1] Scaling Instruction-Finetuned Language Models
>
> [2] The Flan Collection: Designing Data and Methods for Effective Instruction Tuning
>
> $$$$
>
> > `Q4:` Similarly, there is no concrete conclusion from the discussion on reasoning abilities due to insufficient reasoning performance.  One interesting discussion is on the generation order of diffusion LM and its difference from auto-regressive LMs, but a concrete conclusion would probably require a benchmark or a probe dataset for a systematic evaluation, showing the advantage of the generation order employed by diffusion LMs.
>
> Thank you for your interest and valuable suggestions for a more in-depth study on causal generation order. In our paper, we mention two aspects showing promise in eliciting reasoning ability.
> - Qualitatively, we showed the causal generation order of the models, which indicates correct dependency in reasoning.
> - Quantitatively, in Section 4.5, we found the accuracy of our models on GSM8K can attain a non-trivial improvement from 5.6% to 12.8% after tuning on chain-of-thought data [1] of GSM8K distilled from code-davinci-002. This suggests the inability to perform reasoning should be attributed to insufficient model capability instead of the diffusion language modeling paradigm itself.
>
> We believe our findings encourage effort in further scaling and improving the capability of diffusion language models, which facilitates exploring the concrete benefits of causal generation order with the support of sufficiently strong reasoning and planning ability.
>
> [1] Fu et al. Specializing smaller language models towards multi-step reasoning. ICML 2023.

---

> > ### Author Response · Authors · 2023-11-21
> >
> > > `Q5:` The evaluation of diffusion LMs is probably better done with sufficient computing resources for a head-to-head comparison with auto-regressive LMs under similar training conditions.
> >
> > Thanks for your suggestion. We agree that a head-to-head comparison can strictly demonstrate the strengths and weaknesses of diffusion language models against autoregressive language models. However, our work does not aim to build the strongest diffusion language models to beat autoregressive models at this point, but serves as a pioneering investigation to validate the scalability of diffusion language models before consuming huge computational resources. After this validation, as discussed in our future work section, it would be the next step to consider pretraining from scratch.
> >
> > $$$$
> >
> > > `Q6:` How does the performance of diffusion LM compare to other non-autoregressive language models?
> >
> > Most non-autoregressive models struggle to achieve comparable performance as AR.  For example, RDM is among the state-of-the-art non-AR approaches that perform close to AR LM models for supervised tasks like machine translation, whilist our Diffusion LMs perform generally better than other non-AR LMs with the aid of large-scale pretraining. Moreover, we also demonstrate that our large-scale Diffusion LMs can also offer general-purpose generative abilities like AR LMs, which have not been well examined and verified for previous non-autoregressive approaches.
> >
> > $$$$
> >
> > > `Q7:` What about the inference computation cost of diffusion LMs, compared to auto-regressive LMs? How does the number of denoising steps affect the performance-computation tradeoff?
> >
> > **(1) Inference computation cost:**
> > Here are the results of the average latency of generating a sample with `XLM-R-BASE` sized autoregressive and diffusion language models on IWSLT14. The latency of diffusion language models depends on the number of iterations during generation.
> > |                      | Ave. latency (s) ↓ | BLEU  |
> > |----------------------|------------------------------:|-------|
> > | Diffusion (4 iters)  |                          0.05 | 32.20 |
> > | Diffusion (10 iters) |                          0.10 | 35.10 |
> > | Diffusion (25 iters) |                          0.23 | 35.71 |
> > | Autoregressive       |                          0.33 | 33.30 |
> > | Diffusion (50 iters) |                          0.45 | 36.25 |
> >
> >
> > **(2) Performance-computation tradeoff:** Here are the results using different denoising steps of our BASE models finetuned on IWSLT14. Using more sampling steps consistently brings performance improvement while the relative gains diminish.
> > The results indicate that **we can use much fewer sampling steps than 50 to maintain most of the performance**.
> > | # Steps | 1 | 2 |  4 | 10 | 16 | 25 | 35 | 50 | 100 |
> > | --- | --- | --- | --- | --- | --- | --- | --- | --- | --- |
> > | BLEU | 19.65 | 27.95 | 32.20 | 35.10 | 35.37 | 35.71 | 35.48 | 36.25 | 36.22 |
> >
> >
> > $$$$
> >
> >
> > > `Q8:` The performance comparison in Table 1 is a bit confusing: 1) are the RDM models trained by the authors? if so, why not evaluate on all three datasets? 2) "The performance of finetuned XLM-R models is competitive or superior to the common encoder-decoder Transformer baseline." may not be so safe to say as the decoder-only models are larger than encoder-decoder models. 3) how is XLM-R (autoregressive) evaluated on IWSLT14, as it is an encoder-only model?
> >
> > Thank you for your questions and sorry for the confusion. We answer these questions respectfully as follows.
> >
> > 1. We trained these RDM baseline models following previous practice in machine translation and summarization using `Transformer-BASE` architecture on WMT and Gigaword, and using `Transformer-BASE-IWSLT` on IWSLT14. the only difference between `Transformer-BASE` and `Transformer-Base-IWSLT` is the intermediate dimension of FFN (2048 vs 1024).
> > For the missing Gigaword performance with RDM (`Transformer-BASE`), it has a Rouge-L score of 16.25, which is again competitive to an autoregressive counterpart (v.s. 10.42).  We summarize them together as follows.
> > |  | IWSLT14 | WMT14 | Gigaword-10K |
> > | --- | --- | --- | --- |
> > | AR (`Transformer-BASE`) | 33.30  |  26.85 | 10.42 |
> > | RDM (`Transformer-BASE`)  |  32.14 |  26.54 | 16.25 |
> >
> > 2. As you suggested, we conduct a fairer comparison with similar model sizes and configurations. The IWSLT14 performance of the encoder-decoder with the same width and depth as `XLM-R-BASE` is 33.56, similar to the result of 33.30 for `Transformer-BASE-IWSLT14` we reported in Tab. 1. As a result, our conclusion should still hold.
> >
> > 3. The model has the same architecture (i.e., depth, dimension of hidden states, dimension of feedforward layer, embedding size, and number of attention heads) as XLM-R, but the attention mask is the causal mask.

---

> > > ### Author Response · Authors · 2023-11-21
> > >
> > > > `Q9:` In Table 2, why not use zero-shot AR as a reference but use supervised AR?
> > >
> > > Thank you for your suggestion! The supervised AR is a commonly adopted baseline in previous research, with a performance that satisfies practical needs. We include it as a strong reference to provide an intuition that the BLEU score instruction-tuned models achieve is decent. Again, we are not aiming to beat supervised AR in this paper.
> > >
> > > To compare with zero-shot AR LMs as you suggested, we evaluate the zero-shot performance of Flan-T5 and Flan-XLM-R of different sizes on IWSLT14. Both series of models are instruction-tuned on FLAN2022 collections. The results demonstrate that diffusion LMs are also comparable to instruction-tuned autoregressive models like Flan-T5. We will include these results in our next version.
> > > | Diffusion LMs | Flan-XLM-R-Base (86M) | Flan-XLM-R-Large (304M) |  | Flan-XLM-R-XL (2.84B) | Flan-XLM-R-XXL (9.7B) |
> > > | --- | --- | --- | --- | --- | --- |
> > > | BLEU | 21.26 | 25.24 |  | 28.13 | 29.59 |
> > > | **AR LMs** | **Flan-T5-Small (44M)** | **Flan-T5-BASE (198M)** | **Flan-T5-Large (705M)** | **Flan-T5-XL (3B)** | **Flan-T5-XXL (11B)** |
> > > | BLEU | 10.80 | 18.52 | 26.05 | 31.12 | 34.41 |
> > >
> > > $$$$
> > >
> > > > `Q10:` Do the AR models used in the paper use beam search during decoding?
> > >
> > > For task-specific finetuning, we follow standard practice and use beam search with beam size=5. As for instruction tuning, we directly cite the results of Flan-T5 from [1], which does not mention the use of beam search.
> > >
> > > [1] Scaling Instruction-Finetuned Language Models.
> > >
> > >
> > > $$$$
> > >
> > > > `Q11`: Given that diffusion LMs use a different receptive field and generation order than auto-regressive LMs, could there be specific types of tasks or domains where diffusion LMs particularly excel or fall short? That could be a valuable discussion (but probably beyond the scope of the paper).
> > >
> > > Great question! We would like to share some of our insights regarding this.
> > >
> > > We suggest that data that do not conform to a left-to-right generation order will probably benefit from diffusion language models. Typical examples include images, codes, molecules, and protein sequences. In particular, discrete diffusion language models have recently been shown to "beat" typical continuous diffusion models in generating images and videos [1]. Besides, LM-Design [2] has validated the superiority of iterative refinement-based non-autoregressive models in generating protein amino acid sequences given protein structures (a.k.a., protein inverse folding). Given this, diffusion language models should have strong potential in modeling across various modalities beyond natural languages.
> > >
> > > --
> > >
> > > [1] Language Model Beats Diffusion - Tokenizer is key to visual generation. 2023 (under review at ICLR 2024)
> > > > (Note that here in this paper the "language model" is essentially masked LM operating on discrete visual tokens and generating in a mask-predict denoising way, which is conceptually identical to the discrete diffusion LM in our paper)
> > >
> > > [2] Zheng et al. Structure-informed Language Models Are Protein Designers. ICML 2023

---

> ### Author Response · Authors · 2023-11-23
>
> Hello Reviewer xgDh,
>
> As the deadline for author-reviewer discussion is drawing near, we would greatly appreciate receiving your feedback on our responses and are happy to answer further questions.
>
> We look forward to hearing from you, and many thanks!
>
> Sincerely,
> Authors

---

### Official Review · Reviewer_VY65 · 2023-10-31

**Soundness:** 3 good
**Presentation:** 2 fair
**Contribution:** 3 good
**Rating:** 5
**Confidence:** 3

**Summary:**

This paper studies the scalability of discrete diffusion language models on leveraging pretrained knowledge, model scaling and zero to few shot generalizations. The authors first draw an elaborate introduction on diffusion language models. Starting from off-the-shelf pretrained language models, the authors further fine-tune them with diffusion objective following RDM, with either task-specific tuning (MT, Summ) or instruction fine-tuning (FLAN). The authors conduct comprehensive experiments to explore the performance on both down-steam tasks and generalizations of discrete diffusion models varying their scale.

**Strengths:**

- This paper studies an important topic on the scalability of diffusion language models, and goes through extensive experimental explorations, which makes a valuable contribution to developing better diffusion language models.
- This paper poses comprehensive experimental investigations on scaling, tuning and generalizations of discrete diffusion language models, and compares them against the T5 family.
- The author also explores emergent capabilities of autoregressive (L)LMs like in-context learning and reasoning, on their adapted large diffusion language models.
- The paper is well written, clear to follow.

**Weaknesses:**

- The main method in this paper is similar to existing works:

  - The idea of leveraging pretrained masked models as a knowledge base and starting point for tuning discrete diffusion language models (named as "diffusive adaption") is introduced in [1].

  - The main methods in this paper (tuning pretrained MLMs with diffusion objective) is also mainly adhering to the work RDM-Diffusion [2].
  - The "prompt-response" format (partially diffusion on target tokens) essentially adopts the setting of DiffuSeq [3].

- While referring BERT-like diffusion language models as 'decoder-only' models, (GPT-like) AR decoder models (which is currently most popular) is not compared or discussed throughout this study. Besides, I believe that authors need further clarifications on their definitions of 'decoder' models, since we do not generally refer BERT-like models as decoder-only models, which might cause some confusions.

- The paper does not fully explore the configurations of Diffusion LMs, such as the number of diffusion steps, the noise level, and the length predictor. It would be valuable to explore how these factors affect the quality and diversity of the generated texts, especially when the inference latency due to the demand on multiple diffusion steps would limit the scaling of diffusion models. For example, would larger models remedy the demand for more steps & generalize better to predicted lengths or  opposite?

[1] He, Z., Sun, T., Wang, K., Huang, X., & Qiu, X. (2022). Diffusion Bert: Improving generative masked language models with diffusion models. *ArXiv preprint arXiv:2211.15029*.

[2] Zheng, L., Yuan, J., Yu, L., & Kong, L. (2023). A reparametrized discrete diffusion model for text generation. *ArXiv preprint arXiv:2302.05737*.

[3] Gong, S., Li, M., Feng, J., Wu, Z., & Kong, L. (2022, September). DiffuSeq: Sequence to Sequence Text Generation with Diffusion Models. In *The Eleventh International Conference on Learning Representations*.

**Questions:**

- The paper lacks some important baselines and comparisons in Table 1. For example, why are there dashes on most AR transformer baselines? Besides, only small AR models (<100M) are compared against the 9.7B diffusion model. Other important baselines (e.g., GPT-structured decoder models, vanilla AR transformers at similar scale to diffusion variants (~80M), and pretrained + AR fine-tune) should be incorporated to compare against the performance of scaled diffusion models.
- Why are the performance gaps of using oracle length and predicted length opposite in the two MT tasks in Figure 3? Are there any further investigations or explanations?
- On table 2, what is the vanilla performance of XLM-based diffusion models before fine-tuning on FLAN? Since XLM models already adopt a multilingual pretraining objective, it seems natural that it elicits German knowledge and scalable, since larger models undertake greater-scale multilingual pretraining and has more parameters to store this knowledge.
- Why BBH-NLP seems not following the scaling trend in Figure 4?

---

> ### Author Response · Authors · 2023-11-21
>
> We sincerely acknowledge your constructive feedback and suggestions. We would like to address your concerns below. Please have a check and we are happy to address any further feedback!
>
> $$$$
>
> > `Q1:` The main method in this paper is similar to existing works.
>
> Thank you for your question. In this work, we are not only presenting a method; more importantly, we are providing a pioneering exploration into the scalability of diffusion language models, or more broadly, non-autoregressive language models, without the inherent risk of consuming substantial computational resources. We would like to emphasize our additional contributions that go beyond the methods outlined in previous works.
> - DiffusionBERT[1] empirically finds it feasible to tune masked language models into generative models. We further establish the formal connection between reparameterized diffusion models (RDM) and masked language models in terms of training objectives.
> - Furthermore, the application of RDM to advance their methodologies [1,2] results in a commendable boost in competitive performance for this technical routine. This improvement not only facilitates our groundbreaking exploration into scaling diffusion language models but also extends to the broader realm of non-autoregressive language models, all achieved at a significantly lower cost. Our results can inspire further exploration in scaling up non-autoregressive language models, mitigating the previously perceived risk associated with the substantial demand for computational resources.
> - For the "prompt-response" format, we consider it a common practice to apply decoder-only language models for various tasks and do not imply it as a key contribution of our paper.
>
> --
>
> [1] He et al. "Diffusionbert: Improving generative masked language models with diffusion models." 2022
>
> [2] Zheng et al. "Structure-informed Language Models Are Protein Designers". In ICML 2023
>
> $$$$
>
> > `Q2:` The paper lacks some important baselines and comparisons in Table 1. For example, why are there dashes on most AR transformer baselines? Besides, only small AR models (<100M) are compared against the 9.7B diffusion model. Other important baselines (e.g., GPT-structured decoder models, vanilla AR transformers at similar scale to diffusion variants (~80M), and pretrained + AR fine-tune) should be incorporated to compare against the performance of scaled diffusion models.
>
> Thank you for your feedback and suggestions!
>
> **(1) About the dashes on most AR Transformers.**
> Sorry for the confusion. We trained these RDM baseline models following previous practice in machine translation and summarization using Transformer-BASE architecture on WMT and Gigaword, and using Transformer-BASE-IWSLT on IWSLT14.
>
> **(2) About other important baselines you suggested.**
> In Table 1, we have compared a XLM-R-BASE model trained from scratch in both RDM and autoregressive ways. We also conduct vanilla AR Transformer with similar model size with XLM-R-BASE as you suggested.
> To make it clearer, we summarize the results as follows.
> | Model | achitecture | pretrained | IWSLT14 (BLEU) |
> | --- | --- | --- | --- |
> | AR  | vanilla encoder-decoder (~86M) | no | 33.56 |
> | AR (GPT-like) | XLM-R-BASE (decoder-only, ~86M) | no | 26.07 |
> | AR (GPT-like) | XLM-R-BASE (decoder-only, ~86M) | yes | 34.16 |
> | Diffusion | XLM-R-BASE (decoder-only, ~86M) | no | 28.79 |
> | Diffusion | XLM-R-BASE (decoder-only, ~86M) | yes | 34.10 |
>
> As shown in the table, Diffusion LM attains similar performance with encoder-decoder AR LM but largely outperforms decoder-only AR LM with exact same architecture. A notable difference between the models in these two settings lies in the receptive field. Diffusion LM always has a global receptive field on the conditioning input, whereas AR can only perceive the condition with unidirectional attention if not equipped with an encoder. This supports our motivation to build diffusion language models for their favorable global receptive field.
>
> --
>
> [1] Tay et al. "Ul2: Unifying language learning paradigms." In ICLR 2022.
>
> [2] Zhang et al. "Examining scaling and transfer of language model architectures for machine translation." In ICML 2022.

---

> ### Author Response · Authors · 2023-11-21
>
> > `Q3:` While referring BERT-like diffusion language models as 'decoder-only' models, (GPT-like) AR decoder models (which is currently most popular) is not compared or discussed throughout this study. Besides, I believe that authors need further clarifications on their definitions of 'decoder' models, since we do not generally refer BERT-like models as decoder-only models, which might cause some confusions.
>
> Thank you for your questions!
>
> **(1) Clarifications on the definition of the "decoder-only" model.**
> We presented our clarification on the concept of "decoder-only" models in our original submission, that in this paper, the decoder-only architecture, as a counterpart of encoder-decoder architecture, refers to the language model architecture that does not comprise a separate encoder stack to encode contexts/conditions. Under this definition, masked language models (e.g., BERT and XLM-R) are treated as decoder-only models. We will make more clarifications on this in the next version as you suggested.
>
> **(2) Why we used encoder-decoder models (T5) instead of GPT-like decoder-only models?**
> - As for comparisons in Table 1, conventional supervised models on machine translation and summarization are encoder-decoder Transformers, which typically outperform decoder-only Transformers. Table 1 supports this that, on IWSLT14, encoder-decoder AR or Diffusion LMs are consistently much better than decoder-only AR or Diffusion LMs. respectively. As such, we listed them there as strong references to demonstrate how well our model's performance is in comparison to commonly-used supervised baselines.
>
> - Similar reasons go with the comparisons in subsequent experiments regarding instruction-following. We consider T5 as a strong reference model because its representativeness outweighs the mismatch resulting from the architectural differences. As previous work suggests, encoder-decoder models like T5 outperform decoder-only models at moderate scales [1][2][3], and the gap diminishes as models scale up. Besides, the T5 series has both open-sourced pretrained models and instruction-tuned models on the FLAN instruction-tuning dataset at different scales similar to XLM-R. So we believe T5 can serve as a valid and strong baseline for the performance of autoregressive models.
>
> --
>
> [1] Exploring the Limits of Transfer Learning with a Unified Text-to-Text Transformer
>
> [2] UL2: Unifying Language Learning Paradigms
>
> [3] Examining Scaling and Transfer of Language Model Architectures for Machine Translation
>
> [4] Scaling Instruction-Finetuned Language Models
>
> [5] Larger-Scale Transformers for Multilingual Masked Language Modeling
>
>
> $$$$
>
> > `Q4:` The paper does not fully explore the configurations of Diffusion LMs, such as the number of diffusion steps, the noise level, and the length predictor. For example, would larger models remedy the demand for more steps & generalize better to predicted lengths or opposite?
>
> Thank you for your suggestions. We conduct further analyses on these as you suggested.
>
> **(1) Number of diffusion steps.**
> Here are the results using different denoising steps of our models finetuned on IWSLT14. Using more sampling steps consistently brings performance improvement while the relative gains diminish.  The results indicate that we can use much fewer sampling steps than 50 to maintain most of the performance. Additionally, to reach similar performance, larger models require fewer steps (e.g., 10-step LARGE model surpasses the 50-step BASE models).
>
> | # Steps | 1 | 2 |  4 | 10 | 16 | 25 | 35 | 50 |
> | --- | --- | --- | --- | --- | --- | --- | --- | --- |
> | BASE | 19.65 | 27.95 | 32.20 | 35.10 | 35.37 | 35.71 | 35.48 | 36.25 |
> | LARGE | 21.39 | 29.76 | 34.31 | 37.18 | 38.12 | 38.45 | 38.68 | 38.84 |
> | XL | 19.55 | 28.85 | 34.09 | 37.95 | 38.99 | 39.60 | 38.88 | 40.08 |
>
> **(2) About length predictor.**
> We have tried two different parameterizations for length predictor in our preliminary study, either predicting the absolute lengths or predicting the relative length between target responses and their input prompts. We empirically found for machine translation both strategies performed similarly, while for other tasks, absolute length prediction is generally superior. As a result, we decided to use absolute length prediction for all later experiments. Please note that in this study, our focus is not to seek the best configurations but to demonstrate the effectiveness and possibilities of the Diffusion LMs for general-purpose language learning.

---

> > ### Author Response · Authors · 2023-11-21
> >
> > > `Q5:` Why are the performance gaps of using oracle length and predicted length opposite in the two MT tasks in Figure 3? Are there any further investigations or explanations?
> >
> > Generally speaking, results with oracle length are better as more information is provided. But sometimes the predicted length leads to better performance in terms of BLEU score, which is based on n-gram precision, and is biased towards short sentences.
> >
> > $$$$
> >
> > > `Q6:` What is the vanilla performance of XLM-based diffusion models before fine-tuning on FLAN?
> >
> > Before finetuning, the XLM-R models are unable to generate a good full sentence, we suggest this is due to only a constant 15% masking ratio during typical MLM pretraining, which is not compatible with generative denoising sampling where operating on various noise/mask levels is required.
> >
> > We conducted preliminary experiments on this with the 2-shot performances of XLM-R on IWSLT14 De->En before finetuning. The results are shown as follows.
> > |   | base | large | xl | xxl |
> > | --- | --- | --- | --- | --- |
> > | IWSLT14 |  ~0 | 12.12 | 8.91 | 7.78 |
> >
> > Qualitatively, the BASE model fails to follow instructions and generate German while XL and XXL models tend to repeatedly generate irrelevant tokens like emojis. This suggests that the generative ability of masked language models is elicited by our diffusive adaptation.
> >
> > $$$$
> >
> > > `Q7:` Why BBH-NLP seems not following the scaling trend in Figure 4?
> >
> > This is because many questions in BBH-NLP require reasoning to solve. Under the setting of answer-only evaluation, the model performance does not always improve with sizes as demonstrated in [1].
> >
> > --
> >
> > [1] Challenging BIG-Bench Tasks and Whether Chain-of-Thought Can Solve Them

---

> ### Author Response · Authors · 2023-11-23
>
> Dear Reviewer VY65,
>
> As the time for author-reviewer discussion is soon to close, we are looking forward to your valuable feedback on our responses. We are also more than willing to address any further concerns you might have.
>
> Awaiting your response with anticipation and gratitude.
>
>
>
> Best wishes,
>
> The Authors

---

### Official Review · Reviewer_bJbT · 2023-11-02

**Soundness:** 4 excellent
**Presentation:** 3 good
**Contribution:** 2 fair
**Rating:** 5
**Confidence:** 4

**Summary:**

This paper proposes to pre-train a masked language model, and fine-tune this model as a generative discrete absorbing diffusion model for translation, summarization, and instruction-following tasks. Experiments are conducted using the XLM-RoBERTa family of MLMs (86M, 304M, 2.8B, 9.7B variants) finetuned as generative models on IWSLT14 and WMT14 (translation) Gigaword-10K (summarization) and Flan 2022 (instruction-following) datasets.

**Strengths:**

This paper contains a diverse set of experiments using finetuned MLMs as diffusion models, which complements previous work using this methods (e.g., DiffusionBERT). The experiments are described well, and appear to be well-executed. The results show consistent task performance improvements using the larger-scale models, which is consistent with broader observations of the importance of scale in the ML community (although perhaps unsurprising at this point).

**Weaknesses:**

There does not seem to be a significant methodological contribution. As far as I can tell, the methodological approach is the same as previous work adapting pre-trained MLM's for discrete diffusions [1]; the difference here is the use of the XLM-RoBERTa family of models, vs. previous results using BERT. The paper is clear that its contributions are about scale, not methodology, but it's a little frustrating that so much of the paper (up until Section 4) is dedicated to summarizing the contributions of previous work in the field.

This is not the first work to study the scaling behavior of diffusion models for text. For example, multiple diffusion models are trained in [2] together with a scaling law for these models and comparisons to scaling laws for baseline autoregressive models. Because this paper's central contribution is an empirical study of scaling behavior of text diffusions, a more thorough discussion of previous work on scaling text diffusions would be helpful.

[1] DiffusionBERT: Improving Generative Masked Language Models with Diffusion Models
Zhengfu He, Tianxiang Sun, Kuanning Wang, Xuanjing Huang, Xipeng Qiu

[2] Likelihood-Based Diffusion Language Models
Ishaan Gulrajani, Tatsunori B. Hashimoto

**Questions:**

I am curious how the approach described in this paper compares to direct MCMC sampling from an MLM. e.g., the approach described in [3]. To be clear, I am quite sympathetic to the idea of finetuning to directly supervise generation rather than inference-time MCMC; but I do think some discussion of this alternative line of work and connections/tradeoffs vs. the proposed approach would be enlightening and strengthen the discussion in Section 3.2.

"There exist large discrepancies in architecture and data engineering between our foundation models, XLM-R (Conneau et al., 2019), which were built years ago, and the state-of-the-art large language models like LLaMA (Touvron et al., 2023a;b)."

I do not work at a scaling lab, so my insight into these questions is limited, but one explanation I have heard for the lack of recent developments in the MLM space is that the performance of these models simply does not scale well in comparison to autoregressive LM's. I'm curious what you think of that claim, and whether (potential) limitations in our ability to scale the performance of the base MLM could create a ceiling on the performance of diffusion models trained using masked lanuage modeling as a pre-training objective.


[3] Exposing the Implicit Energy Networks behind Masked Language Models via Metropolis--Hastings
Kartik Goyal, Chris Dyer, Taylor Berg-Kirkpatrick

---

> ### Author Response · Authors · 2023-11-21
>
> We sincerely appreciate your reviews! We would like to address your concerns below. Please have a check and we are happy to address any further feedback!
>
> $$$$
>
> > `Q1:` There does not seem to be a significant methodological contribution. As far as I can tell, the methodological approach is the same as previous work adapting pre-trained MLM's for discrete diffusions [1]; the difference here is the use of the XLM-RoBERTa family of models, vs. previous results using BERT.  The paper is clear that its contributions are about scale, not methodology, but it's a little frustrating that so much of the paper (up until Section 4) is dedicated to summarizing the contributions of previous work in the field.
>
> Thanks for your constructive feedback. We would like to also make clarifications on our contributions.
>
> While our work is not the first exploration of generative ability of masked LMs, it is the first to successfully implement and demonstrate the practical effectiveness by scaling these models. The major contribution of our study is therefore to show that the Diffusion LMs can also possess promising generative capabilities that can be elicited from large-scale pretrained MLMs by diffusive adaptation, not only for each individual specialized purpose as in DiffusionBERT, but also generalist abilities like instruction following, in-context learning and promise of reasoning, which are nowadays what the community really cares and excited about LLMs. And this is exactly what we think of something new and surprising, and what we are thrilled to share with the community.
>
> Regarding the content up to Section 4, we believe such summaries are essential. In fact, these contents serve two purposes. Firstly, we try to make necessary concepts clear and easy to follow for audiences from different backgrounds, as there are still limited successful explorations on developing large language models under the probabilistic framework of diffusion models, which are mostly studied in vision domains. Secondly, we try to make it clear (1) how diffusion language models and masked language models are formally connected, and (2) how this connection helps our goal of studying the scaling of diffusion language models.
>
> $$$$
>
> > `Q2:` This is not the first work to study the scaling behavior of diffusion models for text.  For example, multiple diffusion models are trained in [2] together with a scaling law for these models and comparisons to scaling laws for baseline autoregressive models. A more thorough discussion of previous work on scaling text diffusions would be helpful.
>
> While we agree that our work is not the first exploration of scaling diffusion language models, it is the first to explore the techniques of discrete diffusion language models, and successfully demonstrate the practical effectiveness of scaling the models.
> - In detail, the mentioned "Likelihood-Based Diffusion Language Models" paper [2] studies pretraining **continuous** diffusion language models while we explore **discrete** diffusion language models that suit the **discrete nature** of languages better and empirically perform better in language generation benchmarks like machine translation [1].
> - Additionally, [2] scales diffusion language models up to 1B and mainly investigates the scaling law of likelihood estimation (ppl/elbo), with a showcase of the promise of prefix completion and infixing. In comparison, we study Diffusion LMs with scaling up to 10B. More importantly, we benchmark on widely-used LLM testbeds and verify the general-purpose generative abilities of large diffusion language models, which is the crucial property that motivates scaling.
> - We believe that, together with the promising findings in literature [2], our study provide valuable insights into the scalability of diffusion language models and their potential as a viable alternative in tackling generative language tasks across the board.
>
> --
>
> [1] A Reparameterized Discrete Diffusion Model for Text Generation.
>
> [2] Likelihood-Based Diffusion Language Models
>
> $$$$
>
> > `Q3:` How the approach described in this paper compares to direct MCMC sampling from an MLM. I do think some discussion of this alternative line of work and connections/tradeoffs vs. the proposed approach would be enlightening and strengthen the discussion in Section 3.2.
>
> Thank you for your insight question and suggestion. The MCMC sampling methods mentioned adopt Gibbs sampling, which generates one token at a time from left to right. This is similar to autoregressive models. On the contrary, our diffusion models are non-autoregressive and can generate several tokens in parallel without a predefined generation order. In addition, it enables flexible reasoning orders and backtracing as our discussion on reasoning illustrates. We will include the discussion with MCMC sampling as you suggested in our next version.

---

> > ### Author Response · Authors · 2023-11-21
> >
> > > `Q4:` How do you think of the opinion that MLM models do not scale well in comparison to autoregressive LM?
> >
> > We believe the claim that the performance of MLM does not scale well is not true, or at least not well grounded by experiments.
> > - As far as we know, the major limitation of MLMs is that it cannot handle generation tasks well by default, which prevents MLMs from being a general-purpose model. Our work overcomes the limitation of MLM by connecting MLM and diffusion models and to reprogram MLM into Diffusion LM elicit their general-purpose generative ability (i.e., the scaling with many tasks). Moreover, Figure 3 shows our diffusion LMs have close performance to autoregressive models, which suggests they can scale as well as autoregressive models. We consider this as a significant contribution.
> >
> > - Evidence beyond language domains also demonstrate that MLMs can scale well. In particular, the ESM-2 family [1] shows that scaling the size of MLMs on protein sequences ranging from 8M to 15B consistently leads to better results on protein predictive tasks and protein structure prediction. For protein sequence generation, LM-Design further demonstrates that the conditional MLMs can generate better results with model scaling.
> >
> > - In terms of the training infrastructure, MLMs or our diffusion language models can also scale well. Training MLM is almost identical to training a decoder-only autoregressive model, except that they require no causal attention masks. That is, existing infrastructure to train large decoder-only autoregressive models can be immediately used for training large MLM or diffusion language models. (A negative example is encoder-decoder models like T5, whose cross-attention modules make training and inference with pipeline parallelism difficult, thus limiting their scalability from a training infrastructure perspective.)
> >
> > --
> >
> > [1] Lin et al. "Evolutionary-scale prediction of atomic-level protein structure with a language model." Science 2023
> >
> > [2] Zheng et al. "Structure-informed Language Models Are Protein Designers". ICML 2023

---

> ### Author Response · Authors · 2023-11-23
>
> Dear Reviewer bjbT,
>
> With the deadline for author-reviewer discussions approaching, we sincerely appreciate your insightful feedback and are delighted to answer further questions if remain.
>
> We look forward to your reply and thank you for your discussion.
>
> Best,
> The Authors

---

### Official Review · Reviewer_gyd2 · 2023-11-04

**Soundness:** 4 excellent
**Presentation:** 3 good
**Contribution:** 2 fair
**Rating:** 5
**Confidence:** 3

**Summary:**

This work presents an initial exploration into the effects of scale and fine-tuning on large diffusion language models. In particular, the work is interested in answering whether scale and instruction-tuning for large diffusion models can unlock similar generalization capabilities that are observed in more standard, auto-regressive LMs.

To test this, the authors use Reparameterized Discrete Diffusion models (RDM), a method from prior work, which is able to adapt pre-trained masked language models (MLMs) such as BERT into generative diffusion models. The authors use pre-trained XLM-R models, of various sizes, as the backbone of their diffusion models.

The authors use RDM to fine-tune XLM-R on 3 generative NLP tasks (2 translation tasks, and one summarization task). They find that: (a) diffusion models adapted from pre-trained models outperform randomly initialized XLM-R models, suggesting that RDM can leverage pre-training; (b) that performance on all 3 tasks improves with XLM-R backbone size, suggesting that RDM improves with model scale.

The authors next explore the zero-shot capabilities of diffusion LMs after fine-tuning on an instruction-tuning dataset (FLAN), and find that these models are able to achieve relatively strong zero-shot capabilities after instruction-tuning. Finally, the authors explore the ability of these models to perform in-context learning after instruction-tuning, and once again find that diffusion LMs can perform comparably to other in-context LMs after instruction-tuning on certain tasks.

The authors end with an interesting qualitative evaluation of a diffusion LMs reasoning order over a complex reasoning task, showing an example where their diffusion LM generates in an order that holds aligns with a _causal order_, which differs from autoregressive models which must generate a response linearly.

**Strengths:**

- The work demonstrates that diffusion models can be adapted from other pre-trained models, and, when done at scale, can achieve strong performance on certain downstream tasks both via fine-tuning as well as in-context learning.
- The observation that
- The paper is well-written; in particular, it clearly explains prior work on how diffusion LMs operate, what their issues are, and what RDM is.
- The final example, which serves to highlight the differences between diffusion generation and autoregressive generation for chain-of-thought reasoning is very interesting and a novel observation to the best of my knowledge.

**Weaknesses:**

- The finding that pre-training helps an adapted diffusion model, and that pre-trained model size influences the adapted models generalization, is perhaps not surprising. In particular, (a) it is already known XLM-R performance generally improves with scale on several downstream tasks, so the benefit of scale likely comes from the pre-training procedure, rather than the diffusion adaptation and (b) given that pre-training helps over no pre-training, it is not very surprising that _better_ pre-trained models help more.
- The paper claims to be interested in how diffusion LMs scale with respect to data, but that is not tested in this work. What is tested is how diffusion LM capabilities scale w.r.t. pre-trained model size and how diffusion LMs respond to instruction-tuning.
- The final note about how diffusion models generate causally is very interesting, but is only explored with a single qualitative example.
- The results are sometimes a bit confusing to parse.
   - In particular, for Table 1, there is a number of results missing, particularly for XLM-R-BASE (AR), which is perhaps the most comparable portion of the table as it directly compares autoregressive generation to RDM on the exact same model. It's also not clear what the value of having the Encoder-Decoder models there are, since the models are pre-trained differently, have different sizes, and different architectures; the take-away from this comparison is very unclear.
   - I find it a bit surprising that the number of examples used in in-context learning have little effect on model performance (and sometimes have a very negative effect). It would be very useful to have these plotted against a model that is known to perform well at in-context learning, to see if the same trends hold. As it stands, the results suggest to me that the diffusion LMs presented are good zero-shot learners, but poor in-context learners.

**Questions:**

What is the reason for missing fields in Table 1? Are these rows results obtained from other papers? If so, I feel that should be made clearer.

Would it be possible to have Flan-T5 compared to Flan-XLM-R for various numbers of in-context exemplars?

---

> ### Author Response · Authors · 2023-11-21
>
> Thank you so much for the constructive suggestions! We would like to address your concerns below. Please have a check and we are happy to address any further feedback!
>
> $$$$
>
> > `Q1:` It's perhaps not surprising to find the performance of the models improves with model scaling and pretraining. (a) it is already known XLM-R performance generally improves with scale on several downstream tasks, so the benefit of scale likely comes from the pre-training procedure, rather than the diffusion adaptation and (b) given that pre-training helps over no pre-training, it is not very surprising that better pre-trained models help more.
>
> Thank you for your comment! We would like to make clarifications on this as follows:
> - **MLM Pretraining does not permit generative capabilities.** Despite pretraining masked LMs, such as XLM-R, improving downstream tasks with scale, these tasks are generally predictive tasks (regression/classification for NLU) as MLMs are typically thought of as encoders. Whether these pretrained MLMs can solve more tasks in general in a generative manner like causal LMs/GPTs, however, remains under-explored and non-trivial to achieve. Without diffusive adaptation as a generative surgery, only MLM pretraining at scale does not help with generative capabilities on its own.
> - Although it is common to expect that models will perform better with model scaling and pretraining, **it is imperative not to assume this as a given without the backing of empirical results**.
> In particular, previous work on continuous diffusion language models has found it challenging to fit large-scale datasets [1,2,3]. We suggest that our success in scaling diffusion language models is attributed to both adopting the formalization of discrete diffusion models that fit language tokens well and utilizing pretrained masked language models which are well-tested to be scalable.
> - **Scaling is not only about performance improvement but, more importantly, is about eliciting general-purpose ability.** The major contribution of our study is therefore to show that the Diffusion LMs can also possess promising generative capabilities that can be elicited from large-scale pretrained MLMs by diffusive adaptation, not only for each individual specialized purpose as in DiffusionBERT, but also generalist abilities like instruction following, in-context learning and promise of reasoning, which are nowadays what the community really cares and excited about LLMs. And this is exactly what we think of something new and surprising, and what we are thrilled to share with the community.
>
> [1] Yuan et al. "Seqdiffuseq: Text diffusion with encoder-decoder transformers." 2022
>
> [2] Gao et al. "Difformer: Empowering diffusion model on embedding space for text generation." 2022
>
> [3] Ye et al. "Dinoiser: Diffused conditional sequence learning by manipulating noises." 2023
>
>
> > `Q2:` The scaling with respect to data is not tested.
>
> Our purpose is to explore whether pretraining on large-scale data can also empower Diffusion LMs, like their autoregressive counterparts, we mainly focused on diffusive adaptation from pretrained masked LMs (i.e., XLM-Roberta) rather than building Diffusion LMs from scratch for the purpose of highly efficient proof-of-concept. As we have demonstrated in Section 4.2 and Table 1, pretraining on large-scale data gives rise to significant gains for RDM-based Diffusion LMs over the same models without pretraining (e.g., 34.10 vs 28.79 on IWSLT14). We will leave the analysis of the effects of pretraining on different data scales from scratch in the next version.
>
> > `Q3:` The final note about how diffusion models generate causally is very interesting, but is only explored with a single qualitative example.
>
> Thank you for your interest and valuable suggestions for a more in-depth study on causal generation order. In our paper, we mention two aspects showing promise in eliciting reasoning ability.
> - Qualitatively, we showed the causal generation order of the models, which indicates correct dependency in reasoning.
> - Quantitatively, in Section 4.5, we found the accuracy of our models on GSM8K can attain a non-trivial improvement from 5.6% to 12.8% after tuning on chain-of-thought data [1] of GSM8K distilled from `code-davinci-002`. This suggests the inability to perform reasoning should be attributed to insufficient model capability instead of the diffusion language modeling paradigm itself.
>
> We believe our findings encourage effort in further scaling and improving the capability of diffusion language models, which facilitates exploring the concrete benefits of causal generation order with the support of sufficiently strong reasoning and planning ability.
>
> [1] Fu et al. Specializing smaller language models towards multi-step reasoning. ICML 2023.

---

> > ### Author Response · Authors · 2023-11-21
> >
> > > `Q4:` In Table 1, results with XLM-R-BASE (AR) are missing which can directly compare autoregressive generation to RDM on the exact same model. It's also not clear what the value of having the Encoder-Decoder models there are, since the models are pre-trained differently, have different sizes, and different architectures; the take-away from this comparison is very unclear
> >
> > **(1) About autoregressive decoder-only baselines.**
> >
> > Thank you for your suggestion. We actually did these comparisons in Table 1 and explained the findings in footnote 7. However, there may have been some confusion caused by our naming.   In order to make things clearer, we rearrange the results as follows:
> > | models | architecture | IWSLT14 (BLEU) | Gigaword-10K (Rouge-L) |
> > | --- | --- | --- | --- |
> > | AR LM (decoder-only) | XLM-R-BASE | 26.07 | 4.7 |
> > | Diffusion LM | XLM-R-BASE | 28.79 | 10.01 |
> >
> > As shown in the table, Diffusion LM sufficiently outperforms AR LM in the decoder-only setting with exact same architecture. A notable difference between the models in these two settings lies in the receptive field. Diffusion LM always has a global receptive field on the conditioning input, whereas AR can only perceive the condition with unidirectional attention if not equipped with an encoder. This supports our motivation to build diffusion language models for their favorable global receptive field.
> >
> > **(2) About encoder-decoder baselines.**
> >
> > As for comparisons in Table 1, conventional supervised models on machine translation and summarization are encoder-decoder Transformers, which typically outperform decoder-only Transformers. Table 1 supports this that, on IWSLT14, encoder-decoder AR or Diffusion LMs are consistently much better than decoder-only AR or Diffusion LMs. respectively. As such, we listed them there as strong references to demonstrate that how well our model's performance is in comparison to commonly-used supervised baselines.
> >
> > Similar reasons go with the comparisons in subsequent experiments regarding instruction-following. We consider T5 as a strong reference model because its representativeness outweighs the mismatch resulting from the architectural differences. As previous work suggests, encoder-decoder models like T5 outperform decoder-only models at moderate scales [1][2][3], and the gap diminishes as models scale up. Besides, the T5 series has both open-sourced pretrained models and instruction-tuned models on the FLAN instruction-tuning dataset at different scales similar to XLM-R. So we believe T5 can serve as a valid and strong baseline for the performance of autoregressive models.
> >
> > --
> >
> > [1] Exploring the Limits of Transfer Learning with a Unified Text-to-Text Transformer
> >
> > [2] UL2: Unifying Language Learning Paradigms
> >
> > [3] Examining Scaling and Transfer of Language Model Architectures for Machine Translation
> >
> > [4] Scaling Instruction-Finetuned Language Models
> >
> > [5] Larger-Scale Transformers for Multilingual Masked Language Modeling
> >
> > $$$$
> >
> > > `Q5:` The number of examples used in in-context learning has little effect on model performance.  Would it be possible to have Flan-T5 compared to Flan-XLM-R for various numbers of in-context exemplars?
> >
> > This is actually consistent with the findings of the FLAN series paper [1], which shows that model performance is not sensitive to the number of demonstrations after instruction tuning, and we have mentioned this phenomenon in Section 4.2.2. Note that the most important message here is that instruction tuning helps our Diffusion LMs obtain in-context learning ability, without which we found that a raw pretrained MLM cannot learn from demonstrations.
> >
> > As you suggested, we present the few-shot performance of XL sized Flan-T5 and Flan-XLM-R on IWSLT14 here, and will include results on more datasets in our next version. Similar to what we have observed in our manuscript, the performance of both instruction-tuned models changes little even though the number of in-context examples vary.
> > |  | 0-shot | 1-shot | 2-shot |
> > | --- | --- | --- | --- |
> > | Flan-T5-XL (3B) | 31.12 | 31.54 | 31.70 |
> > | Flan-XLM-R-XL (2.89B) | 28.13 | 27.51 | 29.19 |
> >
> > --
> >
> > [1] Scaling Instruction-Finetuned Language Models
> >
> > [2] The Flan Collection: Designing Data and Methods for Effective Instruction Tuning
> >
> > $$$$
> >
> > > `Q6:` What is the reason for missing fields in Table 1? Are these rows results obtained from other papers? If so, I feel that should be made clearer.
> >
> > We are sorry for the confusion. We trained encoder-decoder baseline models following previous practice in machine translation and summarization using Transformer-BASE architecture on WMT and Gigaword, and using Transformer-BASE-IWSLT on IWSLT14. The blankets are because the dataset and architecture mismatches in common practice, although the only difference is the intermediate dimension of FFN (2048 vs 1024)

---

> ### Author Response · Authors · 2023-11-23
>
> Dear Reviewer gyd2,
>
> As the author-reviewer discussion deadline is quickly approaching, we would be very grateful for your valuable feedback on our responses, and be very happy to answer any further questions you may have.
> Looking forward to hearing from you, and many thanks!
>
> Best,
> Authors

---

### Author Response · Authors · 2023-11-21

Dear Reviewers, ACs and SACs,

We want to express sincere appreciation for all reviewers' efforts in reviewing and providing inspiring feedback and valuable suggestions that help us greatly improve our manuscript! We have tried our best to address reviewers' concerns respectively. We will accordingly revise the paper to best include most of the insightful suggestions and comments from the reviewers.

We are very happy to address any further feedback during the discussion phase! And for the reproducibility of model training and the evaluations in our paper, we provide anonymous code here: https://anonymous.4open.science/r/DiffusionLLM_Anonymous-7645/

We do sincerely appreciate you, and cheers!

Authors

---

### Meta-Review · Area_Chair_PkMU · 2023-12-13

**Metareview:**

**Paper Summary:**

This paper investigates the scalability and adaptability of diffusion language models (LMs). Specifically, it explores the adaptation of pretrained masked language models (MLMs) into diffusion LMs and examines their performance across multiple tasks. Utilizing Reparameterized Discrete Diffusion (RDM) models and large XLM-RoBERTa MLMs as backbones, the study finds that diffusion LMs effectively scale with model size and pretraining, demonstrating strong zero-shot and few-shot capabilities after instruction-tuning. Additionally, the paper demonstrates the in-context learning and reasoning abilities of diffusion LMs.

**Strengths:**

1. Extensive Experiments: The paper conducts experiments across various supervised and generative tasks and demonstrates that diffusion models can be adapted from pretrained models, achieving strong performance on downstream tasks through both fine-tuning and in-context learning (gyd2, VY65, xgDh).

**Weaknesses:**

1.  Limited Methodological Contributions: The paper's methodological contributions are limited, primarily building upon existing methods like RDM-Diffusion without significant innovation, and there have been previous works studying the scaling behavior of diffusion LMs (bJbT, VY65, xgDh).
2. Unsurprising Findings: The discovery that pretraining helps, and that performance improves with scale, is not particularly surprising (gyd2).
3. Insufficient Analysis: The exploration into the causal generation order of diffusion models is intriguing but is only supported by a single qualitative example, limiting the strength of this observation (gyd2, xgDh).

**Decision:**

Based on the reviews, while this paper presents extensive experiments on diffusion language models, it falls short in terms of methodological innovation, insights, and detailed analysis of the generation order. Therefore, I am not recommending its acceptance.

**Justification For Why Not Higher Score:**

Based on reviews, this work has limited technological contributions and the findings are not very surprising.

**Justification For Why Not Lower Score:**

N/A

---

### Decision · Program_Chairs · 2024-01-16

Reject